# Diffusion Models for Solving Inverse Problems via Posterior Sampling with Piecewise Guidance

**Saeed Mohseni-Sehdeh**                                    *saeedmohseni@vt.edu*
*Department of Electrical and Computer Engineering*
*Virginia Tech*

**Walid Saad**                                             *walids@vt.edu*
*Department of Electrical and Computer Engineering*
*Virginia Tech*

**Kei Sakaguchi**                              *sakaguchi@mobile.ee.titech.ac.jp*
*Department of Electrical and Electronic Engineering*
*Institute of Science Tokyo*

**Tao Yu**                                      *yutao@mobile.ee.titech.ac.jp*
*Department of Electrical and Electronic Engineering*
*Institute of Science Tokyo*

**Reviewed on OpenReview:** *https://openreview.net/forum?id=nvw3XfvBi7*

## Abstract

Diffusion models are powerful tools for sampling from high-dimensional distributions by progressively transforming pure noise into structured data through a denoising process. When equipped with a guidance mechanism, these models can also generate samples from conditional distributions. In this paper, a novel diffusion-based framework is introduced for solving inverse problems using a piecewise guidance scheme. The guidance term is defined as a piecewise function of the diffusion timestep, facilitating the use of different approximations during high-noise and low-noise phases. This design is shown to effectively balance computational efficiency with the accuracy of the guidance term. Unlike task-specific approaches that require retraining for each problem, the proposed method is problem-agnostic and readily adaptable to a variety of inverse problems. Additionally, it explicitly incorporates measurement noise into the reconstruction process. The effectiveness of the proposed framework is demonstrated through extensive experiments on image restoration tasks, specifically image inpainting and super-resolution. Using a class conditional diffusion model for recovery, compared to the pseudoinverse-guided diffusion model (ΠGDM) baseline, the proposed framework achieves a reduction in inference time of 25% for inpainting with both random and center masks, and 23% and 24% for $4\times$ and $8\times$ super-resolution tasks, respectively, while incurring only negligible loss in PSNR and SSIM.

## 1 Introduction

Diffusion models are a class of deep generative models designed to sample from complex data distributions. Diffusion models have been shown to outperform alternatives like generative adversarial networks (GANs) in image synthesis tasks and currently represent the state of the art in this domain Dhariwal & Nichol (2021). The core idea behind diffusion models is to gradually remove the structure from given input data through a forward diffusion process, transforming it into a tractable distribution, which is typically Gaussian white noise. The model is then trained to learn the reverse process, effectively denoising the data step by step to reconstruct a sample that closely approximates the original data distribution. Diffusion models have been

applied across various fields, including computer vision Baranchuk et al. (2021); Amit et al. (2022), natural language processing Austin et al. (2021); Hoogeboom et al. (2021), audio synthesis Kong et al. (2021), and medical image reconstruction Chung et al. (2023); Cao et al. (2024).

Once trained on a dataset from a specific distribution, diffusion models can generate samples that follow that distribution Yang et al. (2023). These samples are inherently random, as the generation process begins with a noise vector sampled randomly. Consequently, the resulting samples may correspond to any region within the support of the learned distribution. Diffusion models can be employed for conditional sampling Kawar et al. (2022) when their denoising process is adapted to incorporate auxiliary information, enabling the generation of samples that are consistent with the provided conditions.

This conditional sampling capability makes diffusion models promising candidates for solving inverse problems, where the objective is to reconstruct a degraded signal. The core idea is to recover the original signal by sampling from the posterior distribution conditioned on the observed degraded input. In the context of image inverse problems, this approach has demonstrated the ability to produce perceptually high-quality outputs Blau & Michaeli (2018). This motivates the use of diffusion models as effective tools for solving inverse problems.

Numerous challenges arise when employing diffusion models to solve inverse problems. These models are typically large and require extensive computational resources and substantial amounts of data for effective training. Given the wide variety of inverse problems, it is impractical to train a separate diffusion model for each specific task Song et al. (2022a); Kawar et al. (2022); Song et al. (2022b). Therefore, a key challenge is developing a unified framework that can address multiple inverse problems using a single, pre-trained diffusion model without the need for task-specific retraining. Another key challenge is maintaining high-quality restoration, typically measured by standard metrics such as the peak signal-to-noise ratio (PSNR) and structural similarity index (SSIM). A general-purpose system must achieve results that are competitive with models trained specifically for individual inverse problems. In addition, computational efficiency is critical: The faster the conditional sampling process, the more practical the framework becomes.

## 1.1 Related Works

Various deep neural network-based techniques have been proposed for solving inverse problems Ongie et al. (2020); Venkatakrishnan et al.; Romano et al. (2017); Mataev et al. (2019); Bora et al. (2017); Daras et al. (2021); Menon et al. (2020) and the aforementioned challenges. These methods can be broadly categorized into supervised (problem-specific) and unsupervised (problem-agnostic) approaches. In supervised methods, the degradation model is known during both training and inference. In contrast, unsupervised methods assume that the degradation is only known at inference time. The unsupervised approaches are particularly appealing, as they better reflect real-world scenarios where access to degradation models during training is often limited or unavailable, and these approaches do not rely on training problem-specific models.

One class of unsupervised deep neural network techniques addresses inverse problems by iteratively applying a pretrained model Venkatakrishnan et al.; Romano et al. (2017); Mataev et al. (2019); Sun et al. (2019). Methods such as plug-and-play (PnP) Venkatakrishnan et al., regularization by denoising (RED) Romano et al. (2017), and their successors in Mataev et al. (2019) and Sun et al. (2019) incorporate a denoiser into an iterative recovery process. Another line of work leverages (GANs) Bora et al. (2017); Daras et al. (2021); Menon et al. (2020), where the latent space of a pretrained GAN is searched to find latent codes that generate images best aligned with the observed measurements. These methods often require a large number of iterations to converge to a satisfactory solution making them time inefficient and not practical for real use cases in which low inference time is critical.

Diffusion models are another class of deep neural networks that can be used for solving inverse problems, with applications in both supervised Kadkhodaie & Simoncelli (2021); Jalal et al. (2021); Kawar et al. (2021a;b) and unsupervised Chung et al. (2022); Dhariwal & Nichol (2021); Saharia et al. (2022; 2023) settings. Denoising diffusion reconstruction models (DDRM) Kawar et al. (2022) represent a diffusion-based approach for solving unsupervised inverse problems. In this method, denoising is performed in the spectral domain of the degradation matrix, and the results are subsequently transformed back into the original image space. While DDRM has demonstrated promising restoration quality, its effectiveness is limited in scenarios

where the relationship between the measurement noise level and the diffusion noise level in the spectral domain is weak.

Another diffusion-based method for unsupervised inverse problems is the pseudoinverse-guided diffusion model (ΠGDM) Song et al. (2022a). This approach computes the guidance term using a one-step denoising approximation of the posterior distribution of the data conditioned on the noisy latent diffusion variable. Unlike DDRM, ΠGDM enables updates regardless of the levels of diffusion and measurement noise. Although effective, ΠGDM requires computing the derivative of the denoiser's output with respect to its input, a computationally intensive operation, particularly when the denoiser's complexity and the dimensionality of the data increase.

The computational and time complexity of GAN-based methods and ΠGDM, the sensitivity of DDRM to measurement noise, and the need for fine-tuning and retraining in supervised approaches highlight the need for a new, reliable, problem-agnostic framework for solving inverse problems. Such a framework should deliver high-quality restoration with low inference time and computational cost, while also explicitly accounting for measurement noise, which is almost always present in real-world scenarios.

## 1.2 Contributions

The main contribution of this paper is the introduction of a new diffusion-based framework for solving inverse problems, which uses a *piecewise function* to approximate the guidance term. This approach preserves accuracy while significantly reducing computational complexity. Moreover, the proposed method explicitly accounts for measurement noise. Specifically, our key contributions include:

- We propose a novel, problem-agnostic, diffusion-based framework for solving inverse problems via posterior sampling, which employs a *piecewise guidance function* that depends on both the measurement (with possible additive measurement noise) and the noisy latent variable at each diffusion time step.

- We show how the proposed method leverages the varying noise and information content of latent variables across time steps to compute *time-dependent guidance* values, enabling a more effective tradeoff between computational efficiency and reconstruction accuracy.

- We derive mathematical expressions that quantify the quality of the approximation in terms of the Kullback–Leibler (KL) divergence between the true and approximated distributions used in the guidance computation, providing insights into how problem parameters affect the effectiveness of the approximation.

- Extensive experiments show that the proposed method reduces inference time while maintaining comparable performance in terms of PSNR and SSIM. Compared to ΠGDM, it achieves a reduction of 25% for both inpainting with random and center masks, and 23% and 24% for $4\times$ and $8\times$ super-resolution, respectively, using a class conditional diffusion model.

The rest of this paper is organized as follows. Section 2 introduces the problem and its challenges. Then, Section 3 details the proposed method. Section 4 presents a theoretical analysis of how the problem parameters influence the proposed solution, and Section 5 presents and analyzes the simulation results. Finally, conclusions are drawn in Section 6.

## 2 Problem Statement

Inverse problems are a class of signal processing problems whose objective is to recover an original signal from observed measurements. Consider the following model that relates an original signal $\boldsymbol{x}_0 \in \mathbb{R}^n$ with a corresponding measurement $\boldsymbol{y} \in \mathbb{R}^m$ of that signal:

$$\boldsymbol{y} = \boldsymbol{C}\boldsymbol{x}_0 + \boldsymbol{z}, \tag{1}$$

where $\boldsymbol{C} \in \mathbb{R}^{m \times n}$ represents the measurement model and $\boldsymbol{z} \in \mathbb{R}^m$ is an i.i.d measurement noise distributed according to $\mathcal{N}(\boldsymbol{0}, \sigma_z^2 \boldsymbol{I})$. In practical inverse problems $m < n$. The objective is to recover $\boldsymbol{x}_0$ from the measurement $\boldsymbol{y}$. Inverse problems are central to domains such as medical imaging Bertero et al. (2021), computer vision Mohammad-Djafari et al. (2023) (e.g., image deblurring, super-resolution), and astronomy Craig & Brown (1986).

Inverse problems are inherently ill-posed Calvetti & Somersalo (2018); small changes in the observations can lead to large variations in the solution. Moreover, solutions to inverse problems may not be unique Calvetti & Somersalo (2018), requiring the incorporation of prior knowledge and regularization to obtain meaningful results. This "ill-posedness" and non-uniqueness arise from the indirect, incomplete, and noisy nature of the measurements. In (1), the incompleteness of the measurement is reflected in the condition $m < n$.

Different approaches have been proposed for solving inverse problems, such as regularization-based techniques, which impose constraints on the solution, such as sparsity or low-rank structure Bertero et al. (2021), and statistical approaches, which assume a prior distribution on the original signal and solve the problem by finding suitable estimators Bertero et al. (2021). One method that has been shown to produce high-quality solutions is *posterior sampling* Blau & Michaeli (2018). The core idea of this approach is to solve (1) by generating samples from the conditional distribution $p(\boldsymbol{x}_0|\boldsymbol{y})$. Diffusion models can be used for sampling from such distributions, provided their denoising procedures are appropriately modified.

Diffusion models perform sampling from distributions by first structurally removing data information and increasing noise. This is achieved by forming a chain of random variables which progressively get noisier. This procedure is called *forward path* and it continues until the point that the structure in the data is reduced to pure noise, where sampling from that is practical. Diffusion models by learning the reverse process of removing noise gradually from the noisy data known as *backward path*, achieve sampling from the distribution.

When the chain of created noisy random variables modeled as a continuum, the forward path of diffusion models can be expressed by a stochastic differential equation (SDE) of the form

$$d\boldsymbol{x} = \boldsymbol{f}(\boldsymbol{x}, t)dt + g(t)d\boldsymbol{w}_f, \tag{2}$$

where $\boldsymbol{f} : \mathbb{R}^{n+1} \to \mathbb{R}^n$ is the drift term which represents the deterministic trend of the state variable $\boldsymbol{x}$ and slowly removes the presence of data, $g : \mathbb{R} \to \mathbb{R}$ represents the intensity of the randomness affecting the state variable and gradually increases the presence of noise in the data and $\boldsymbol{w}_f$ is standard Wiener processes. One such choice that has been adopted heavily are $\boldsymbol{f}(\boldsymbol{x}, t) = -\frac{1}{2}\beta(t)\boldsymbol{x}$ and $g(t) = \sqrt{\beta(t)}$, in which $\beta(t) : \mathbb{R} \to (0, 1)$ is a monotonically increasing function of $t$.

As shown in Anderson (1982), (2) can be solved in reverse by solving

$$d\boldsymbol{x} = [\boldsymbol{f}(\boldsymbol{x}, t) - \frac{1}{2}g(t)^2 \nabla_{\boldsymbol{x}} \log p_t(\boldsymbol{x})] + g(t)d\boldsymbol{w}_b, \tag{3}$$

where the time variable $t$ moves in the opposite direction of (2), $\boldsymbol{w}_b$ is the standard Wiener process in reverse time, and $p_t$ is the probability distribution function (PDF) of the random variable $\boldsymbol{x}(t)$. As shown in Anderson (1982), (2) and (3) are equivalent in the sense that the PDF of $\boldsymbol{x}(t)$ is the same for the solutions of both for all $t$. Thus, solving (3) can result in samples from $p_t(\boldsymbol{x})$ for each time step $t$, and $t = 0$ corresponds to the PDF of data. In practice, (3) is solved using numerical methods such as Euler–Maruyama, provided that $\nabla_{\boldsymbol{x}} \log p_t(\boldsymbol{x})$ (the score function) is known for all the noise levels, which can be approximated with using score estimating models. In numerical solvers the continuous process $\boldsymbol{x}(t)$ is discretized as $\boldsymbol{x}(t) = \boldsymbol{x}_t$ where $t \in \{1, 2, \ldots, T\}$. Then, (3) is solved by assuming that $\boldsymbol{x}_T$ is almost pure noise, and a random sample from that serves as the starting point of the numerical solvers.

In order to use diffusion models for solving inverse problems using posterior sampling, one can solve (3) with replacing $\nabla_{\boldsymbol{x}} \log p_t(\boldsymbol{x})$ with its problem specific counterpart, $\nabla_{\boldsymbol{x}_t} \log p_t(\boldsymbol{x}_t|\boldsymbol{y})$, where $p_t(\boldsymbol{x}_t|\boldsymbol{y})$ represents the PDF of the noisy variable at time step $t$ given the measurement. To calculate the problem-specific score function, Bayes' rule can be used

$$\nabla_{\boldsymbol{x}_t} \log p_t(\boldsymbol{x}_t|\boldsymbol{y}) = \nabla_{\boldsymbol{x}_t} \log p_t(\boldsymbol{x}_t) + \nabla_{\boldsymbol{x}_t} \log p_t(\boldsymbol{y}|\boldsymbol{x}_t), \tag{4}$$

where $\nabla_{\boldsymbol{x}_t} \log p_t(\boldsymbol{x}_t)$ can be approximated using score estimation networks trained on the original data and it is part of the diffusion model training process. The second term (guidance term) in (4) adapts the sampling procedure to produce samples that are consistent with the given measurement.

Unfortunately, the calculation of a closed-form expression for $\nabla_{\boldsymbol{x}_t} \log p_t(\boldsymbol{y}|\boldsymbol{x}_t)$ is generally intractable. The reason for this is the presence of the diffusion noise on both sides of $p_t(\boldsymbol{y}|\boldsymbol{x}_t)$. To elaborate, consider the following equation derived from (2)

$$\boldsymbol{x}_t = \sqrt{\bar{\alpha}_t}\boldsymbol{x}_0 + \sqrt{1 - \bar{\alpha}_t}\boldsymbol{\epsilon}, \tag{5}$$

where $\bar{\alpha}_t = \prod_{i=1}^{t} \alpha_t$, $\alpha_t = 1 - \beta_t$ and $\beta_t = \beta(t)$ for $t \in \{1, 2, \ldots, T\}$ and $\boldsymbol{\epsilon} \sim \mathcal{N}(\boldsymbol{0}, \boldsymbol{I})$. Then, for $p_t(\boldsymbol{y}|\boldsymbol{x}_t)$ we have

$$p_t(\boldsymbol{y}|\boldsymbol{x}_t) = p(\boldsymbol{y} = \boldsymbol{C}\boldsymbol{x}_0 + \boldsymbol{z}|\boldsymbol{x}_t = \sqrt{\bar{\alpha}_t}\boldsymbol{x}_0 + \sqrt{1 - \bar{\alpha}_t}\boldsymbol{\epsilon})$$
$$= p\left(\boldsymbol{y} = \frac{\boldsymbol{C}\boldsymbol{x}_t}{\sqrt{\bar{\alpha}_t}} - \frac{\sqrt{1 - \bar{\alpha}_t}}{\sqrt{\bar{\alpha}_t}}\boldsymbol{C}\boldsymbol{\epsilon} + \boldsymbol{z}|\boldsymbol{x}_t\right). \tag{6}$$

The presence of diffusion noise $\boldsymbol{\epsilon}$ in both the conditioned and conditioning parts of (6) makes the calculation intractable.

The computation of the problem-specific score function is crucial to the performance of diffusion-based methods for solving inverse problems. For the approach to be practical, the score function must be computable without training a separate model, as model retraining for each new problem reduces generality and increases computational overhead. Moreover, the computation should be efficient to avoid compromising inference speed and must account for measurement noise inherent in the observations. To address these challenges, we propose a method that enables efficient problem-specific score computation that accounts for measurement noise without model retraining. The next section outlines the key components of our approach.

## 3 Proposed Conditional Score Estimation

We propose to approximate $p_t(\boldsymbol{y}|\boldsymbol{x}_t)$ by using a piecewise function that varies across different time steps along the diffusion path. The idea is that (6) can be simplified if $\frac{\sqrt{1 - \bar{\alpha}_t}}{\sqrt{\bar{\alpha}_t}}\boldsymbol{C}\boldsymbol{\epsilon} = \boldsymbol{0}$. There is no control over the value of $\boldsymbol{C}\boldsymbol{\epsilon}$ as the diffusion noise is random and $\boldsymbol{C}$ is fixed, thus the only way to make the expression small is through the coefficient $(\frac{\sqrt{1 - \bar{\alpha}_t}}{\sqrt{\bar{\alpha}_t}})$. By design of the diffusion models, this coefficient is always small at lower time steps $t$, near the end of the backward path as for those values we have $\bar{\alpha}_t \approx 1$. This condition is a property of diffusion models and does not depend on the inverse problem, making it a versatile property. Under this observation, for low values of $t$ in the diffusion path, $p_t(\boldsymbol{y}|\boldsymbol{x}_t)$ can be computed as follows

$$p_t(\boldsymbol{y}|\boldsymbol{x}_t) = p\left(\boldsymbol{y} = \frac{1}{\sqrt{\bar{\alpha}_t}}\boldsymbol{C}\boldsymbol{x}_t - \frac{\sqrt{1 - \bar{\alpha}_t}}{\sqrt{\bar{\alpha}_t}}\boldsymbol{C}\boldsymbol{\epsilon} + \boldsymbol{z}|\boldsymbol{x}_t\right),$$
$$\approx p\left(\boldsymbol{z} = \boldsymbol{y} - \frac{1}{\sqrt{\bar{\alpha}_t}}\boldsymbol{C}\boldsymbol{x}_t|\boldsymbol{x}_t\right) \qquad t < T_0,$$
$$= p\left(\boldsymbol{z} = \boldsymbol{y} - \frac{1}{\sqrt{\bar{\alpha}_t}}\boldsymbol{C}\boldsymbol{x}_t\right) = \mathcal{N}\left(\boldsymbol{y}; \frac{1}{\sqrt{\bar{\alpha}_t}}\boldsymbol{C}\boldsymbol{x}_t, \sigma_z^2\boldsymbol{I}\right),$$

where $T_0$ is the highest time step at which this approximation performs well. The final expression follows from the assumption that the measurement noise $\boldsymbol{z}$ and the diffusion noise $\boldsymbol{\epsilon}$ are independent, which is a reasonable assumption because measurement plays no role in the creation of the diffusion noise. With this assumption, the score function will be

$$\nabla_{\boldsymbol{x}_t} \log p_t(\boldsymbol{y}|\boldsymbol{x}_t) = \frac{1}{\sigma_z^2 \sqrt{\bar{\alpha}_t}}\boldsymbol{C}^T\left(\boldsymbol{y} - \frac{1}{\sqrt{\bar{\alpha}_t}}\boldsymbol{C}\boldsymbol{x}_t\right) \qquad t < T_0. \tag{7}$$

In order to compute the guidance term for $T > t > T_0$, we use the approach in Song et al. (2022a) in which the guidance term is approximated with one-step denoising. To elaborate, note that the following equation

holds:

$$p_t(\boldsymbol{y}|\boldsymbol{x}_t) = \int p(\boldsymbol{y}|\boldsymbol{x}_0, \boldsymbol{x}_t)p(\boldsymbol{x}_0|\boldsymbol{x}_t)d\boldsymbol{x}_0 \tag{8}$$

$$= \int p(\boldsymbol{y}|\boldsymbol{x}_0)p(\boldsymbol{x}_0|\boldsymbol{x}_t)d\boldsymbol{x}_0.$$

As suggested in Song et al. (2022a), $p(\boldsymbol{x}_0|\boldsymbol{x}_t)$ can be approximated by the one step denoising that estimates $\boldsymbol{x}_0$ from $\boldsymbol{x}_t$ and to account for the randomness in denoising process, $p(\boldsymbol{x}_0|\boldsymbol{x}_t)$ is modeled as a Gaussian distribution around the estimated $\boldsymbol{x}_0$. In other words

$$p(\boldsymbol{x}_0|\boldsymbol{x}_t) \sim \mathcal{N}(\hat{\boldsymbol{x}}_0^t, r_t^2\boldsymbol{I}), \tag{9}$$

where $\hat{\boldsymbol{x}}_0^t$ is the estimate of the $\boldsymbol{x}_0$ from the given $\boldsymbol{x}_t$ obtained from the denoising mechanism of the diffusion models and $r_t$ controls the variance of the approximate distribution. Using (8) and (9), it can be shown that

$$p_t(\boldsymbol{y}|\boldsymbol{x}_t) \sim \mathcal{N}(\boldsymbol{C}\hat{\boldsymbol{x}}_0^t, r_t^2\boldsymbol{C}\boldsymbol{C}^T + \sigma_z^2\boldsymbol{I}). \tag{10}$$

The score function then follows as

$$\nabla_{\boldsymbol{x}_t} \log p_t(\boldsymbol{y}|\boldsymbol{x}_t) \approx (\frac{\partial \hat{\boldsymbol{x}}_0^t}{\partial \boldsymbol{x}_t})^T \boldsymbol{C}^T (r_t^2\boldsymbol{C}\boldsymbol{C}^T + \sigma_z^2\boldsymbol{I})^{-1}(\boldsymbol{y} - \boldsymbol{C}\hat{\boldsymbol{x}}_0^t). \tag{11}$$

Putting this altogether, the proposed guidance term will be:

$$\nabla_{\boldsymbol{x}_t} \log p_t(\boldsymbol{x}_t|\boldsymbol{y}) = \nabla_{\boldsymbol{x}_t} \log p_t(\boldsymbol{x}_t) \tag{12}$$

$$+ \begin{cases} \frac{1}{\sigma_z^2 \sqrt{\bar{\alpha}_t}}\boldsymbol{C}^T(\boldsymbol{y} - \frac{1}{\sqrt{\bar{\alpha}_t}}\boldsymbol{C}\boldsymbol{x}_t) & \text{if } t < T_0, \\ (\frac{\partial \hat{\boldsymbol{x}}_0^t}{\partial \boldsymbol{x}_t})^T \boldsymbol{C}^T (r_t^2\boldsymbol{C}\boldsymbol{C}^T + \sigma_z^2\boldsymbol{I})^{-1}(\boldsymbol{y} - \boldsymbol{C}\hat{\boldsymbol{x}}_0^t), & \text{o.w,} \end{cases} \tag{13}$$

where $r_t$ is selected as in Song et al. (2022a)

Algorithm 1 outlines the complete step-by-step implementation of the proposed method. It reconstructs the original data from the given measurement matrix $\boldsymbol{C}$ and the observed measurement $\boldsymbol{y}$. The key distinction between this approach and ΠGDM lies in the use of a conditional statement that enables piecewise guidance based on the diffusion time step. Unlike ΠGDM, which uses the same formula to compute the problem-specific score function across all diffusion time steps, the proposed method employs a computationally simpler function at lower time steps, leading to accelerated inference at those time steps.

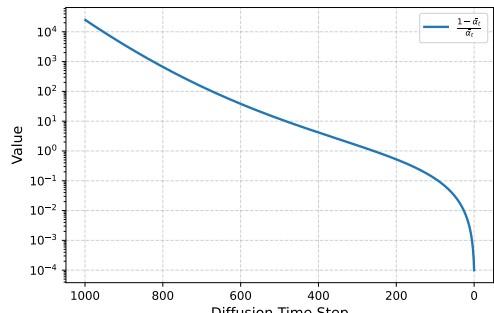

Figure 1: Behavior of the coefficient $\frac{1-\bar{\alpha}_t}{\bar{\alpha}_t}$ across diffusion time steps, shown on a logarithmic scale. As characterized by Theorems 1 and 2, the rapid decay of this coefficient at lower diffusion time steps reduces the KL divergence between the true and approximated problem-specific score functions, thereby supporting the validity of the approximation in this regime.

The computational advantage of using a piecewise guidance function can be understood by observing (11), which involves the computation of a vector-Jacobian product. This requires taking derivatives of the denoiser's output with respect to its input, a computationally expensive operation. Additionally, this expression includes the inversion of a matrix, which further increases the computational cost. In contrast to Song et al. (2022a), which uses these computationally expensive operations across all the diffusion steps, our piecewise guidance approach substitutes such operations with the simpler form given in (7) for certain time steps, reducing the computational cost. Using a piecewise function to approximate the score function takes advantage of the different levels of diffusion noise at each time step to accelerate sampling in the final diffusion steps. It offers a reduction in inference time while maintaining performance. The proposed method requires specifying the parameter $T_0$. Next, we provide preliminary insights into how $T_0$ influences performance, offering an intuitive understanding of its selection.

---

**Algorithm 1** Posterior sampling for inverse problems via piecewise guidance

---

**Inputs:** $\boldsymbol{y} \in \mathbb{R}^m$, $\boldsymbol{C} \in \mathbb{R}^{m \times n}$, $\sigma_z$, $\eta \in [0, 1]$, diffusion-based noise predictor model $\boldsymbol{D}$, $r_t$ (estimation noise sequence) $T_0$, $k_1$ and $k_2$ (for tuning).
**Initialize:** $\boldsymbol{x} \sim \mathcal{N}(\boldsymbol{0}, \boldsymbol{I})$
**for** $t = N$ to 1 **do**
  $\hat{\boldsymbol{\epsilon}} \leftarrow \boldsymbol{D}(\boldsymbol{x}, t)$  $\triangleright$ Predicts the added noise at time step $t$
  $\hat{\boldsymbol{x}} \leftarrow \frac{\boldsymbol{x} - \sqrt{1 - \bar{\alpha}_t}}{\sqrt{\bar{\alpha}_t}}$
  $c_1 \leftarrow \eta \sqrt{(1 - \frac{\bar{\alpha}_t}{\bar{\alpha}_{t-1}}) \frac{1 - \bar{\alpha}_{t-1}}{1 - \bar{\alpha}_t}}$  $\triangleright$ $c_1, c_2$ : DDIM coeficients
  $c_2 \leftarrow \sqrt{1 - \bar{\alpha}_{t-1} - c_1^2}$
    **if** $i < T_0$ **then**
      $\boldsymbol{g} \leftarrow k_1 \frac{1}{\sigma_z^2 \sqrt{\bar{\alpha}_t}} \boldsymbol{C}^T (\boldsymbol{y} - \boldsymbol{C} \frac{\boldsymbol{x}}{\sqrt{\bar{\alpha}_t}})$
    **else**
      $\boldsymbol{g} \leftarrow k_2 (\frac{\partial \hat{\boldsymbol{x}}}{\partial \boldsymbol{x}})^T \boldsymbol{C}^T (r_t^2 \boldsymbol{C} \boldsymbol{C}^T + \sigma_z^2 \boldsymbol{I})^{-1} (\boldsymbol{y} - \boldsymbol{C} \hat{\boldsymbol{x}})$
    **end if**
  $\boldsymbol{\epsilon} \sim \mathcal{N}(\boldsymbol{0}, \boldsymbol{I})$
  $\boldsymbol{x} \leftarrow \sqrt{\bar{\alpha}_{t-1}} + c_1 \boldsymbol{\epsilon} + c_2 \hat{\boldsymbol{\epsilon}} + \sqrt{\bar{\alpha}_t} \boldsymbol{g}$
**end for**

---

Table 1: High-level comparison of the computational operations required to compute the guidance term.

| Method | Steps $t > T_0$ | Steps $t \leq T_0$ |
|---|---|---|
| ΠGDM | matrix multiplication + backpropagation | matrix multiplication + backpropagation |
| Ours | matrix multiplication + backpropagation | matrix multiplication |

## 4 Theoretical Analysis of Piecewise Score Estimation

In this section, we present a theoretical analysis of the proposed method by examining the relationship between the true conditional distribution under the idealized setting of knowing the diffusion noise at each step and its proposed approximation at lower time steps. We derive explicit expressions for the differences between these distributions, as well as for the differences between their corresponding guidance terms. This analysis provides insight into the approximation's effectiveness, identifies the factors influencing its accuracy, and offers a quantitative basis for selecting $T_0$.

To formalize this, we begin by considering the Kullback–Leibler (KL) divergence between the noise conditioned true distribution $p_0(\boldsymbol{y}|\boldsymbol{x}_t, \boldsymbol{v}_t)$ and its proposed approximation $p_a(\boldsymbol{y}|\boldsymbol{x}_t)$ under the assumption that the noise term becomes negligible for lower time steps, i.e., $\frac{\sqrt{1-\bar{\alpha}_t}}{\sqrt{\bar{\alpha}_t}} \boldsymbol{C} \boldsymbol{\epsilon} \approx \boldsymbol{0} \; \forall t < T_0$. We additionally analyze the deviation between the true and approximated guidance vectors, resulting from these distributions. The following theorem provides a closed-form expression for these quantities.

**Theorem 1.** Suppose the value of the added noise for the latent diffusion variable at time step $t$ is $\boldsymbol{v}_t \in \mathbb{R}^n$, i.e., $\boldsymbol{x}_t = \sqrt{\bar{\alpha}_t} \boldsymbol{x}_0 + \sqrt{1 - \bar{\alpha}_t} \boldsymbol{v}_t$, then the value of the KL divergence between the noise conditioned true distribution $p_0(\boldsymbol{y}|\boldsymbol{x}_t, \boldsymbol{v}_t)$ and the approximated version $p_a(\boldsymbol{y}|\boldsymbol{x}_t)$ is $\frac{1}{2\sigma_z^2} \frac{1-\bar{\alpha}_t}{\bar{\alpha}_t} \|\boldsymbol{C} \boldsymbol{v}_t\|_2^2$. Further more the $\ell_2$-norm difference between the guidance terms corresponding to these distributions is $\frac{\sqrt{1-\bar{\alpha}_t}}{\bar{\alpha}_t} \|\boldsymbol{C}^T \boldsymbol{C} \boldsymbol{v}_t\|_2$.

*Proof.* To prove Theorem 1, we need the following lemma from Hershey & Olsen (2007):

**Lemma 1.** If $\boldsymbol{X}_1 \sim \mathcal{N}(\boldsymbol{\mu}_1, \boldsymbol{\Sigma}_1) \in \mathbb{R}^n$ and $\boldsymbol{X}_2 \sim \mathcal{N}(\boldsymbol{\mu}_2, \boldsymbol{\Sigma}_2) \in \mathbb{R}^n$, then the KL divergence between the distribution of these two variables is given by

$$\frac{1}{2} \left[ \log \frac{|\boldsymbol{\Sigma}_2|}{|\boldsymbol{\Sigma}_1|} - n + \text{tr}(\boldsymbol{\Sigma}_2^{-1} \boldsymbol{\Sigma}_1) + (\boldsymbol{\mu}_2 - \boldsymbol{\mu}_1)^T \boldsymbol{\Sigma}_2^{-1} (\boldsymbol{\mu}_2 - \boldsymbol{\mu}_1) \right].$$

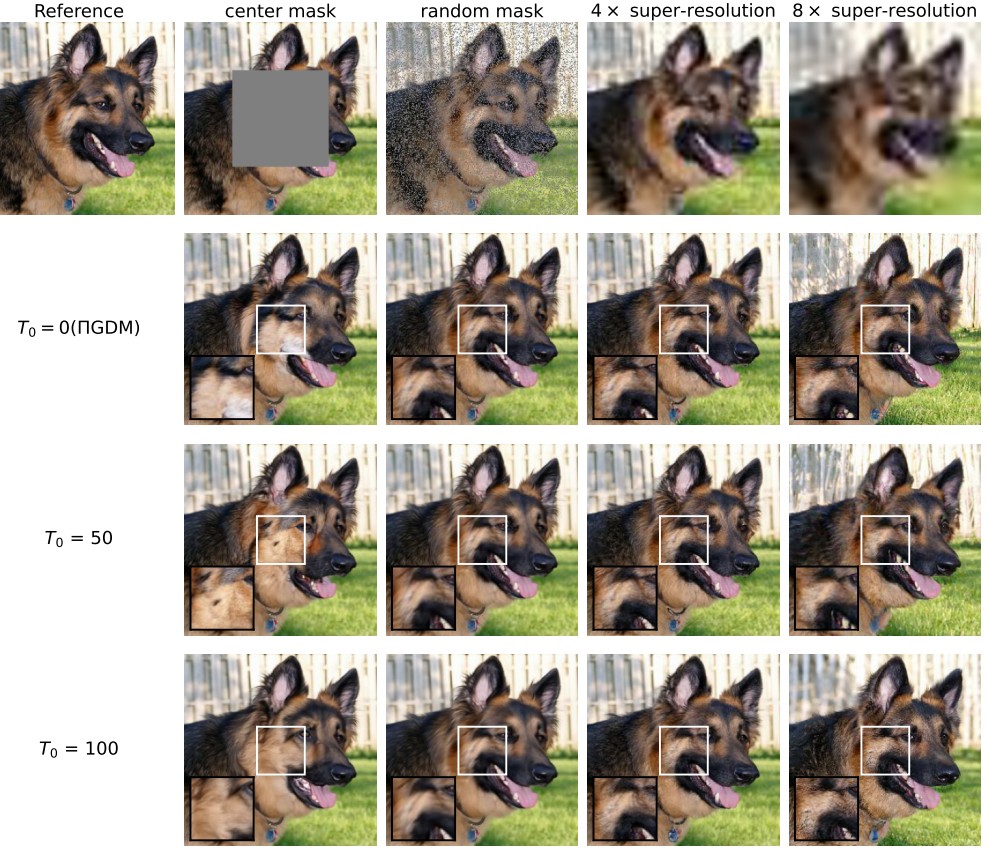

Figure 2: Restoration results on four inverse problems, inpainting with center mask (second column from left), inpainting with random mask, covering 30% of the pixels (third column form left), 4× super-resolution (fourth column from left) and 8× super-resolution (the fifth column from left) for different values of the $T_0$ using an image of a dog using **class conditional** diffusion model. $T_0$ controls when each component of the piece-wise guidance term is active. ΠGDM corresponds to $T_0 = 0$.

The noise conditioned true distribution $p_0(\boldsymbol{y}|\boldsymbol{x}_t, \boldsymbol{v}_t)$ can be written as

$$p_0(\boldsymbol{y}|\boldsymbol{x}_t, \boldsymbol{v}_t) = p(\boldsymbol{y} = \boldsymbol{C}\boldsymbol{x}_0 + \boldsymbol{z}|\boldsymbol{x}_t = \sqrt{\bar{\alpha}_t}\boldsymbol{x}_0 + \sqrt{1-\bar{\alpha}_t}\boldsymbol{v}_t) = p\left(\boldsymbol{z} = \boldsymbol{y} - \frac{1}{\sqrt{\bar{\alpha}_t}}\boldsymbol{C}\boldsymbol{x}_t + \frac{\sqrt{1-\bar{\alpha}_t}}{\sqrt{\bar{\alpha}_t}}\boldsymbol{C}\boldsymbol{v}_t\right)$$

$$= \mathcal{N}\left(-\frac{\sqrt{1-\bar{\alpha}_t}}{\sqrt{\bar{\alpha}_t}}\boldsymbol{C}\boldsymbol{v}_t + \frac{1}{\sqrt{\bar{\alpha}_t}}\boldsymbol{C}\boldsymbol{x}_t, \sigma_z^2\boldsymbol{I}\right),$$

where the first line follows from (2) and (1), the second line follows from the assumption that at each time step $t$, the corresponding true noise $\boldsymbol{v}_t$ is known and the third line follows from the model assumption that the measurement noise follows $\mathcal{N}(\boldsymbol{0}, \boldsymbol{I})$. For the proposed method, by the assumption of $\frac{\sqrt{1-\bar{\alpha}_t}}{\sqrt{\bar{\alpha}_t}}\boldsymbol{C}\boldsymbol{v}_t = \boldsymbol{0}$ for low values of $t$, the approximate conditional distribution $p_a(\boldsymbol{y}|\boldsymbol{x}_t)$ can be written as

$$p_a(\boldsymbol{y}|\boldsymbol{x}_t) = p(\boldsymbol{y} = \boldsymbol{C}\boldsymbol{x}_0 + \boldsymbol{z}|\boldsymbol{x}_t = \sqrt{\bar{\alpha}_t}\boldsymbol{x}_0 + \sqrt{1-\bar{\alpha}_t}\boldsymbol{v}_t)$$

$$= p\left(\boldsymbol{z} = \boldsymbol{y} - \frac{1}{\sqrt{\bar{\alpha}_t}}\boldsymbol{C}\boldsymbol{x}_t\right) = \mathcal{N}\left(\frac{1}{\sqrt{\bar{\alpha}_t}}\boldsymbol{C}\boldsymbol{x}_t, \sigma_z^2\boldsymbol{I}\right).$$

Using Lemma 1, we have

$$D_{\mathrm{KL}}(p_0(\boldsymbol{y}|\boldsymbol{x}_t, \boldsymbol{v}_t)\,\|\,p_a(\boldsymbol{y}|\boldsymbol{x}_t)) = \frac{1}{2\sigma_z^2}\frac{1-\bar{\alpha}_t}{\bar{\alpha}_t}\|\boldsymbol{C}\boldsymbol{v}_t\|_2^2. \tag{14}$$

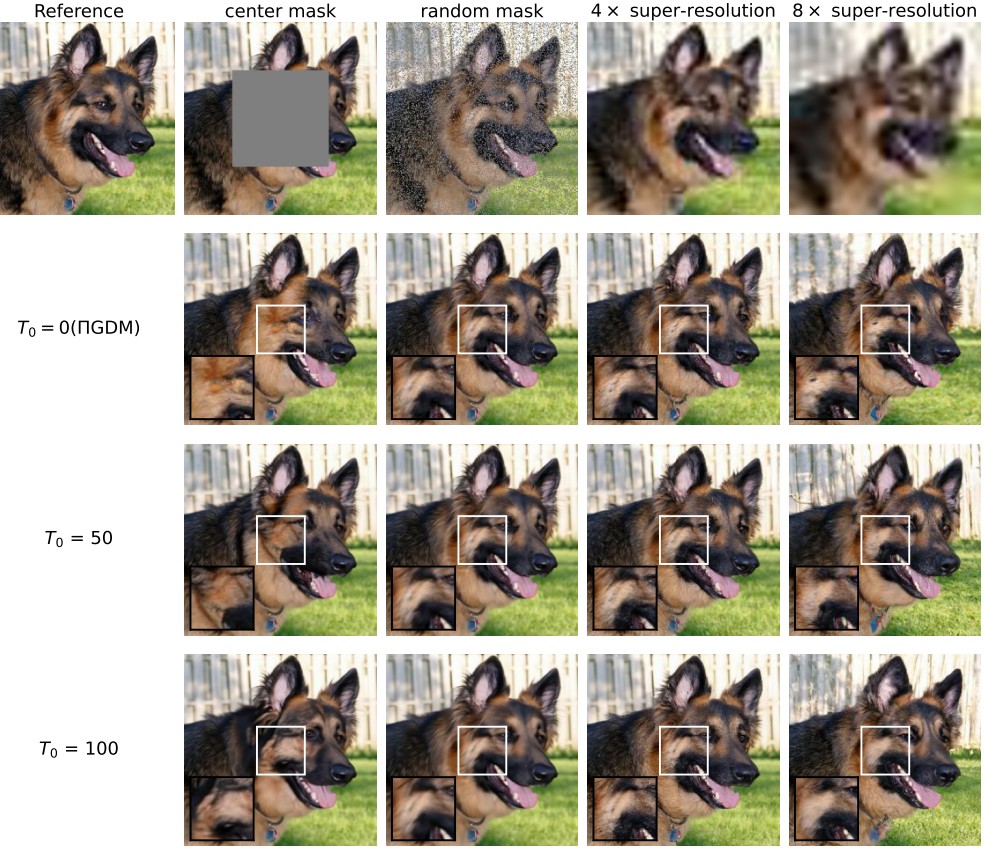

Figure 3: Restoration results on four inverse problems, inpainting with center mask (second column from left), inpainting with random mask, covering 30% of the pixels (third column form left), 4× super-resolution (fourth column from left) and 8× super-resolution (the fifth column from left) for different values of the $T_0$ using an image of a dog using **class unconditional** diffusion model. $T_0$ controls when each component of the piece-wise guidance term is active. ΠGDM corresponds to $T_0 = 0$.

To compute the $\ell_2$ norm of the difference between the guidance terms, recall that the guidance term is defined as $\nabla_{\boldsymbol{x}_t} \log p(\boldsymbol{y}|\boldsymbol{x}_t)$. The $\ell_2$ norm of the difference between the guidance computed under the ideal setting, where the diffusion noise is known at each diffusion step and approximation is then given by

$$d = \|\nabla_{\boldsymbol{x}_t} \log p_0(\boldsymbol{y}|\boldsymbol{x}_t, \boldsymbol{v}_t) - \nabla_{\boldsymbol{x}_t} \log p_a(\boldsymbol{y}|\boldsymbol{x}_t)\|_2 \tag{15}$$

Since $p_0(\boldsymbol{y}|\boldsymbol{x}_t, \boldsymbol{v}_t) = \mathcal{N}\left(-\frac{\sqrt{1-\bar{\alpha}_t}}{\sqrt{\bar{\alpha}_t}}\boldsymbol{C}\boldsymbol{v}_t + \frac{1}{\sqrt{\bar{\alpha}_t}}\boldsymbol{C}\boldsymbol{x}_t, \sigma_z^2\boldsymbol{I}\right)$, and $p_a(\boldsymbol{y}|\boldsymbol{x}_t) = \mathcal{N}\left(\frac{1}{\sqrt{\bar{\alpha}_t}}\boldsymbol{C}\boldsymbol{x}_t, \sigma_z^2\boldsymbol{I}\right)$, we have

$$d = \|-\frac{1}{\sqrt{\bar{\alpha}_t}}\boldsymbol{C}^T\left(\boldsymbol{y} - \frac{1}{\sqrt{\bar{\alpha}_t}}\boldsymbol{C}\boldsymbol{x}_t + \frac{\sqrt{1-\bar{\alpha}_t}}{\sqrt{\bar{\alpha}_t}}\boldsymbol{C}\boldsymbol{v}_t\right) + \frac{1}{\sqrt{\bar{\alpha}_t}}\boldsymbol{C}^T\left(\boldsymbol{y} - \frac{1}{\sqrt{\bar{\alpha}_t}}\boldsymbol{C}\boldsymbol{x}_t\right)\|_2 \tag{16}$$

$$= \|-\frac{1}{\sqrt{\bar{\alpha}_t}}\boldsymbol{C}^T\left(\frac{\sqrt{1-\bar{\alpha}_t}}{\sqrt{\bar{\alpha}_t}}\boldsymbol{C}\boldsymbol{v}_t\right)\|_2 = \frac{\sqrt{1-\bar{\alpha}_t}}{\bar{\alpha}_t}\|\boldsymbol{C}^T\boldsymbol{C}\boldsymbol{v}_t\|_2$$

$\square$

This theorem provides insight into the performance of the proposed approximation. According to Theorem 1, the KL divergence between the approximate and the true noise conditional distributions, as well as the difference between the corresponding guidance terms at each time step $t$, are influenced by the parameter $\bar{\alpha}_t$ and the degradation matrix $\boldsymbol{C}$. When $\bar{\alpha}_t$ is close to one, these quantities become negligible, indicating

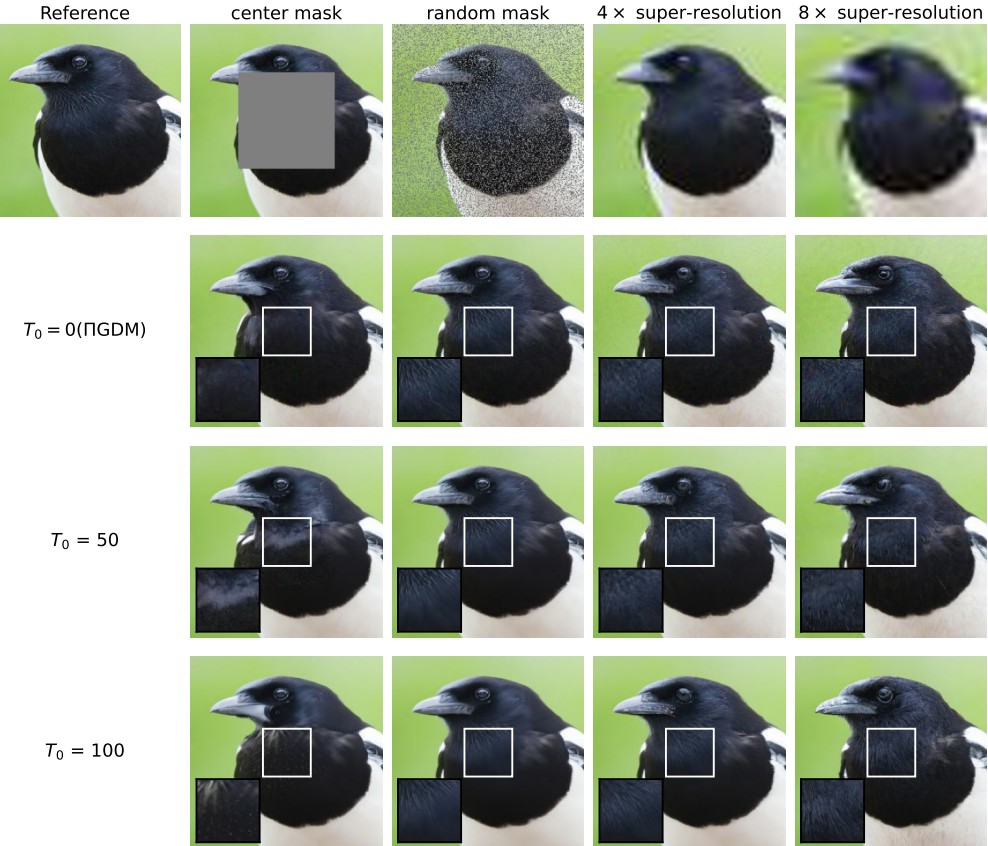

Figure 4: Restoration results on four inverse problems, inpainting with center mask (second column from left), inpainting with random mask, covering 30% of the pixels (third column form left), $4\times$ super-resolution (fourth column from left) and $8\times$ super-resolution (the fifth column from left) for different values of the $T_0$ using an image of a magpie using **class conditional** diffusion model. $T_0$ controls when each component of the piece-wise guidance term is active. $\Pi$GDM corresponds to $T_0 = 0$.

that the approximation closely matches the true distribution. Furthermore, (14) and (16) highlight the role of the degradation matrix $\boldsymbol{C}$ in determining the quality of the approximation, as it directly affects both the KL divergence and the guidance difference.

Next, we derive an expression for the KL divergence between the true noise conditioned distribution $p_0(\boldsymbol{y}|\boldsymbol{x}_t, \boldsymbol{v}_t)$ and its approximation $p_{a'}(\boldsymbol{y}|\boldsymbol{x}_t)$, as well as for the deviation between their corresponding guidance vectors. The approximate distribution is obtained by estimating the added noise using a denoiser.

**Theorem 2.** Suppose that the added noise for the latent diffusion variable at time step $t$ is denoted by $\boldsymbol{v}_t \in \mathbb{R}^n$, i.e., $\boldsymbol{x}_t = \sqrt{\bar{\alpha}_t}\boldsymbol{x}_0 + \sqrt{1 - \bar{\alpha}_t}\boldsymbol{v}_t$. Let $\hat{\boldsymbol{v}}_t = \boldsymbol{D}(\boldsymbol{x}_t, t)$ be the estimate of the true noise $\boldsymbol{v}_t$, obtained by the denoiser $\boldsymbol{D}$ from $\boldsymbol{x}_t$ and the time step $t$. Assume that the denoiser output at each time step $t$ is centered around the true noise with an additive zero-mean random error, i.e., $\hat{\boldsymbol{v}}_t = \boldsymbol{v}_t + \boldsymbol{\epsilon}_t$, where $\mathbb{E}[\boldsymbol{\epsilon}_t] = \boldsymbol{0}$. Then, the KL divergence between the true noise conditioned distribution $p_0(\boldsymbol{y}|\boldsymbol{x}_t, \boldsymbol{v}_t)$ and its approximation $p_{a'}(\boldsymbol{y}|\boldsymbol{x}_t)$ is given by $\frac{1}{2\sigma_z^2}\frac{1-\bar{\alpha}_t}{\bar{\alpha}_t}\|\boldsymbol{C}\boldsymbol{\epsilon}_t\|_2^2$. Furthermore, the $\ell_2$-norm difference between the guidance terms corresponding to these distributions is $\frac{\sqrt{1-\bar{\alpha}_t}}{\bar{\alpha}_t}\|\boldsymbol{C}^T\boldsymbol{C}\boldsymbol{\epsilon}_t\|_2$.

*Proof.* Similar to the proof of Theorem 1, for the true conditional distribution we have

$$p_0(\boldsymbol{y}|\boldsymbol{x}_t, \boldsymbol{v}_t) = \mathcal{N}\left(-\frac{\sqrt{1-\bar{\alpha}_t}}{\sqrt{\bar{\alpha}_t}}\boldsymbol{C}\boldsymbol{v}_t + \frac{1}{\sqrt{\bar{\alpha}_t}}\boldsymbol{C}\boldsymbol{x}_t, \sigma_z^2\boldsymbol{I}\right). \tag{17}$$

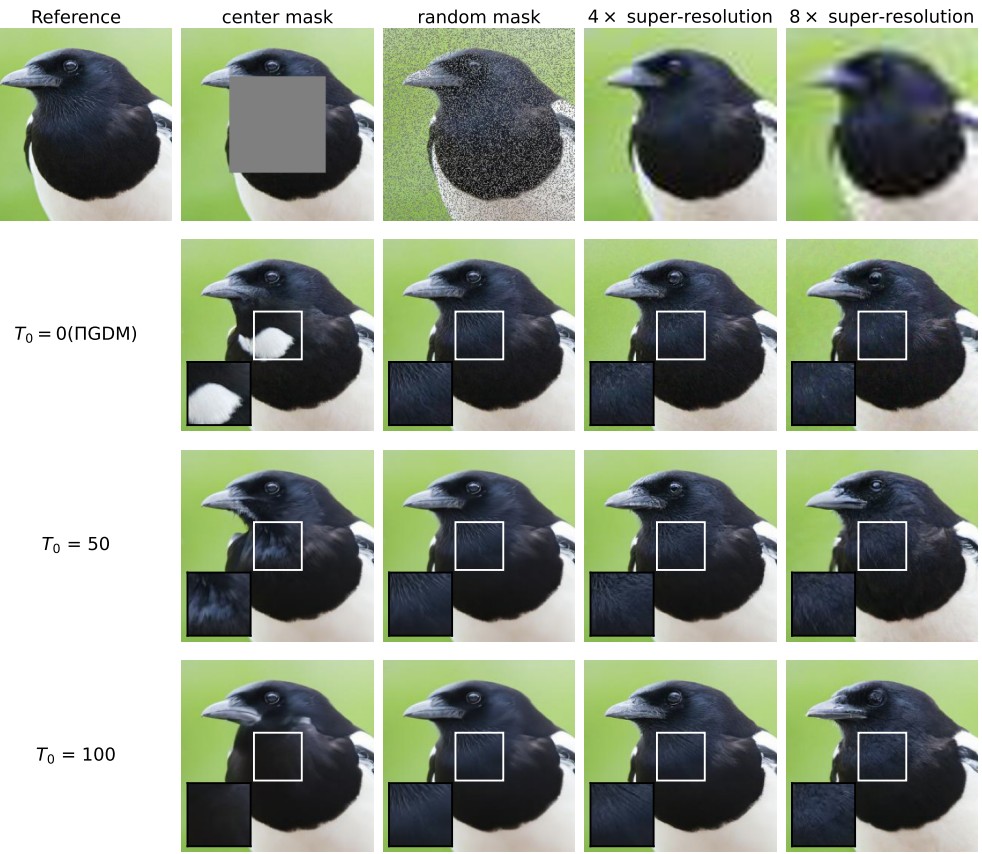

Figure 5: Restoration results on four inverse problems, inpainting with center mask (second column from left), inpainting with random mask, covering 30% of the pixels (third column form left), $4\times$ super-resolution (fourth column from left) and $8\times$ super-resolution (the fifth column from left) for different values of the $T_0$ using an image of a magpie using **class unconditional** diffusion model. $T_0$ controls when each component of the piece-wise guidance term is active. $\Pi$GDM corresponds to $T_0 = 0$.

For the approximate distribution $p_{a\prime}(\boldsymbol{y}|\boldsymbol{x}_t)$, we obtain

$$p_{a\prime}(\boldsymbol{y}|\boldsymbol{x}_t) = p\left(\boldsymbol{z} = \boldsymbol{y} - \frac{1}{\sqrt{\bar{\alpha}_t}}\boldsymbol{C}\boldsymbol{x}_t + \frac{\sqrt{1-\bar{\alpha}_t}}{\sqrt{\bar{\alpha}_t}}\boldsymbol{C}\hat{\boldsymbol{v}}_t\right) = \mathcal{N}\left(-\frac{\sqrt{1-\bar{\alpha}_t}}{\sqrt{\bar{\alpha}_t}}\boldsymbol{C}\hat{\boldsymbol{v}}_t + \frac{1}{\sqrt{\bar{\alpha}_t}}\boldsymbol{C}\boldsymbol{x}_t, \sigma_z^2\boldsymbol{I}\right). \tag{18}$$

Then, by applying Lemma 1, we obtain

$$D_{\mathrm{KL}}(p_0(\boldsymbol{y}|\boldsymbol{x}_t, \boldsymbol{v}_t) \,\|\, p_{a\prime}(\boldsymbol{y}|\boldsymbol{x}_t)) = \frac{1}{2\sigma_z^2}\frac{1-\bar{\alpha}_t}{\bar{\alpha}_t}\|\boldsymbol{C}\boldsymbol{\epsilon}_t\|_2^2. \tag{19}$$

To compute the $\ell_2$ norm of the difference between the corresponding guidance terms, we proceed analogously to Theorem 1:

$$\begin{aligned} d &= \|\nabla_{\boldsymbol{x}_t} \log p_0(\boldsymbol{y}|\boldsymbol{x}_t, \boldsymbol{v}_t) - \nabla_{\boldsymbol{x}_t} \log p_{a\prime}(\boldsymbol{y}|\boldsymbol{x}_t)\|_2 \\ &= \|-\frac{1}{\sqrt{\bar{\alpha}_t}}\boldsymbol{C}^T\left(\boldsymbol{y} - \frac{1}{\sqrt{\bar{\alpha}_t}}\boldsymbol{C}\boldsymbol{x}_t + \frac{\sqrt{1-\bar{\alpha}_t}}{\sqrt{\bar{\alpha}_t}}\boldsymbol{C}\boldsymbol{v}_t\right) + \frac{1}{\sqrt{\bar{\alpha}_t}}\boldsymbol{C}^T\left(\boldsymbol{y} - \frac{1}{\sqrt{\bar{\alpha}_t}}\boldsymbol{C}\boldsymbol{x}_t + \frac{\sqrt{1-\bar{\alpha}_t}}{\sqrt{\bar{\alpha}_t}}\boldsymbol{C}\hat{\boldsymbol{v}}_t\right)\|_2 \\ &= \frac{\sqrt{1-\bar{\alpha}_t}}{\bar{\alpha}_t}\|\boldsymbol{C}^T\boldsymbol{C}(\hat{\boldsymbol{v}}_t - \boldsymbol{v}_t)\|_2 = \frac{\sqrt{1-\bar{\alpha}_t}}{\bar{\alpha}_t}\|\boldsymbol{C}^T\boldsymbol{C}\boldsymbol{\epsilon}_t\|_2. \end{aligned} \tag{20}$$

$\square$

Theorem 2 shows that the KL divergence and the $\ell_2$-norm of the difference between the guidance terms can again remain small for lower values of $t$, since $\bar{\alpha}_t \approx 1$ in those time steps. However, depending on its performance, employing a denoiser can provide improved accuracy at the cost of increased computational effort. Specifically, according to Theorem 2, the terms $\|\boldsymbol{C}\boldsymbol{\epsilon}_t\|_2^2$ and $\|\boldsymbol{C}^T\boldsymbol{C}\boldsymbol{\epsilon}_t\|_2$ decrease when the denoiser performs well and its estimation error $\boldsymbol{\epsilon}_t$ is close to zero. Nonetheless, this noise estimation step increases the computational complexity of the denoising process.

Theorem 1 provides the foundation for analyzing the choice of the parameter $T_0$, which determines when each component of the piecewise function becomes active. The following theorem offers a quantitative criterion for selecting $T_0$ based on the tolerance of the error in the guidance term difference computed from the true noise conditioned distribution and it's approximate obtained under the assumption that $\frac{\sqrt{1-\bar{\alpha}_t}}{\sqrt{\bar{\alpha}_t}}\boldsymbol{C}\boldsymbol{v}_t = \boldsymbol{0}$ for $t \leq T_0$.

**Theorem 3.** Suppose that the added noise for the latent diffusion variable at time step $t$ is denoted by $\boldsymbol{v}_t \in \mathbb{R}^n$, i.e., $\boldsymbol{x}_t = \sqrt{\bar{\alpha}_t}\boldsymbol{x}_0 + \sqrt{1-\bar{\alpha}_t}\boldsymbol{v}_t$. Let $p_0(\boldsymbol{y}|\boldsymbol{x}_t, \boldsymbol{v}_t)$ denote the true noise conditioned distribution and $p_a(\boldsymbol{y}|\boldsymbol{x}_t)$ its approximate counterpart, under the assumption that $\frac{\sqrt{1-\bar{\alpha}_t}}{\sqrt{\bar{\alpha}_t}}\boldsymbol{C}\boldsymbol{v}_t = \boldsymbol{0}$. Then, for all time steps $t \leq T_0$ satisfying $\bar{\alpha}_{T_0} \geq \frac{-1+\sqrt{1+4\delta}}{2\delta}$, where $\delta = \frac{\epsilon^2}{\|\boldsymbol{C}^T\boldsymbol{C}\|_F}$, the expected $\ell_2$-norm of the difference between the guidance terms corresponding to the true and approximate conditional distributions is bounded by $\epsilon$; that is,

$$\mathbb{E}[d] = \mathbb{E}\left[\|\nabla_{\boldsymbol{x}_t}\log p_0(\boldsymbol{y}|\boldsymbol{x}_t, \boldsymbol{v}_t) - \nabla_{\boldsymbol{x}_t}\log p_a(\boldsymbol{y}|\boldsymbol{x}_t)\|_2\right] \leq \epsilon.$$

*Proof.* To prove this theorem, by using the result of Theorem 1, we have

$$\mathbb{E}[d^2] = \frac{1-\bar{\alpha}_t}{\bar{\alpha}_t^2}\mathbb{E}\left[\|\boldsymbol{C}^T\boldsymbol{C}\boldsymbol{v}_t\|_2^2\right] = \frac{1-\bar{\alpha}_t}{\bar{\alpha}_t^2}\mathbb{E}\left[\boldsymbol{v}_t^T(\boldsymbol{C}^T\boldsymbol{C})^2\boldsymbol{v}_t\right]. \tag{21}$$

Since, by the structure of the forward process, we have $\boldsymbol{v}_t \sim \mathcal{N}(\boldsymbol{0}, \boldsymbol{I})$, (21) simplifies to

$$\mathbb{E}\left[\boldsymbol{v}_t^T(\boldsymbol{C}^T\boldsymbol{C})^2\boldsymbol{v}_t\right] = \text{tr}\left((\boldsymbol{C}^T\boldsymbol{C})^2\boldsymbol{I}\right) = \|\boldsymbol{C}^T\boldsymbol{C}\|_F^2. \tag{22}$$

Hence, by using Jensen's inequality we obtain

$$\mathbb{E}[d] \leq \sqrt{\mathbb{E}[d^2]} = \frac{\sqrt{1-\bar{\alpha}_t}}{\bar{\alpha}_t}\|\boldsymbol{C}^T\boldsymbol{C}\|_F. \tag{23}$$

The condition $\mathbb{E}[d] \leq \epsilon$ is guaranteed if $\frac{\sqrt{1-\bar{\alpha}_t}}{\bar{\alpha}_t}\|\boldsymbol{C}^T\boldsymbol{C}\|_F \leq \epsilon$. Solving this inequality for $\bar{\alpha}_t$, we find that it holds when

$$\bar{\alpha}_t \geq \frac{-1+\sqrt{1+4\delta}}{2\delta}, \quad \text{where} \quad \delta = \frac{\epsilon^2}{\|\boldsymbol{C}^T\boldsymbol{C}\|_F^2}.$$

Since $\bar{\alpha}_t$ decreases with respect to $t$, the parameter $T_0$ can be chosen as the first time step for which $\bar{\alpha}_t \geq \frac{-1+\sqrt{1+4\delta}}{2\delta}$ holds, ensuring the condition is satisfied for all $t \leq T_0$. $\qquad\square$

Theorem 3 provides a quantitative criterion for selecting the parameter $T_0$ given a prescribed tolerance $\delta$. This tolerance reflects the allowable error in the guidance term for the inverse problem and depends on the desired score function estimation accuracy $\epsilon$ as well as on the problem-specific degradation operator through the quantity $\|\boldsymbol{C}^\top\boldsymbol{C}\|_F$. As $\delta$ increases for a $\boldsymbol{C}$, the admissible value of $T_0$ increases accordingly, corresponding to a regime in which larger approximation errors in the score function are acceptable. This, in turn, allows for larger choices of $T_0$ and consequently reduces the computational cost. The theorem thus provides a principled starting point for selecting an appropriate $T_0$ once the degradation matrix of the problem $\boldsymbol{C}$ and the tolerated guidance error are specified.

## 5 Simulation Results and Analysis

We evaluate the effectiveness of the proposed method on two inverse problems: Image super-resolution and image inpainting. All simulations are conducted on $256 \times 256$ images in alignment with standard settings adopted in prior work e.g., Kawar et al. (2022) and Song et al. (2022a).

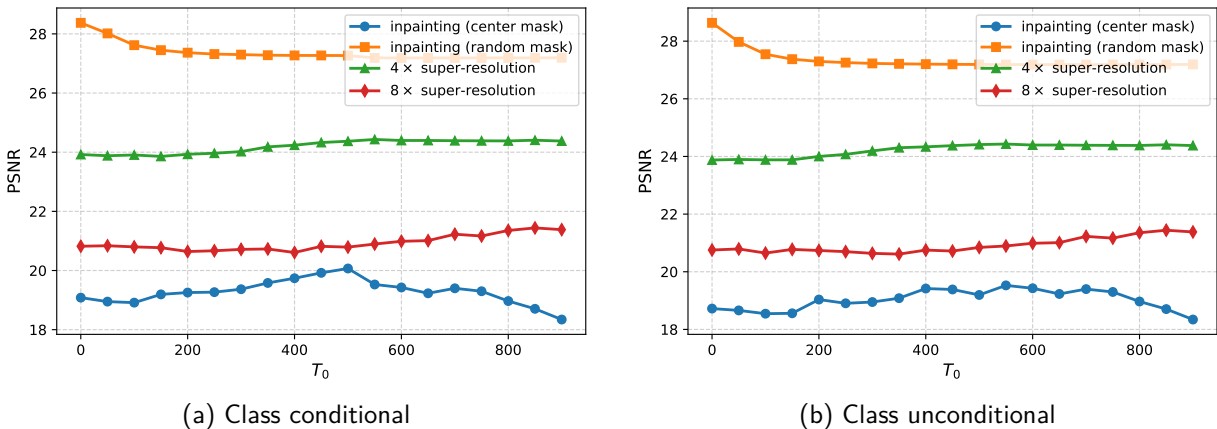

(a) Class conditional  (b) Class unconditional

Figure 6: Average PSNR scores across various inverse problems as a function of the guidance threshold $T_0$, which controls when each component of the piece-wise guidance term is active. $\Pi$GDM corresponds to $T_0 = 0$.

For these simulations, we primarily use a subset of 50 images selected from the ImageNet dataset Russakovsky et al. (2015), which is widely used in image generation and restoration tasks. Unless otherwise stated, all reported results are obtained on this 50-image subset. For a specific experiment, we additionally evaluate the method on a larger subset of 1000 ImageNet images to assess scalability and robustness. All the following simulations are done using an NVIDIA H200 GPU (140 GB) on a server with AMD EPYC 7742 CPU.

For the diffusion model, we rely on a publicly available pretrained diffusion model[1] Dhariwal & Nichol (2021), trained specifically on $256 \times 256$ images. The model follows a standard denoising schedule with 1000 diffusion steps. Our method integrates this model into an inverse problem framework, and we compare its performance with $\Pi$GDM Song et al. (2022a), a recently proposed problem-agnostic approach that has demonstrated strong empirical performance across various inverse problems, often rivaling task-specific methods.

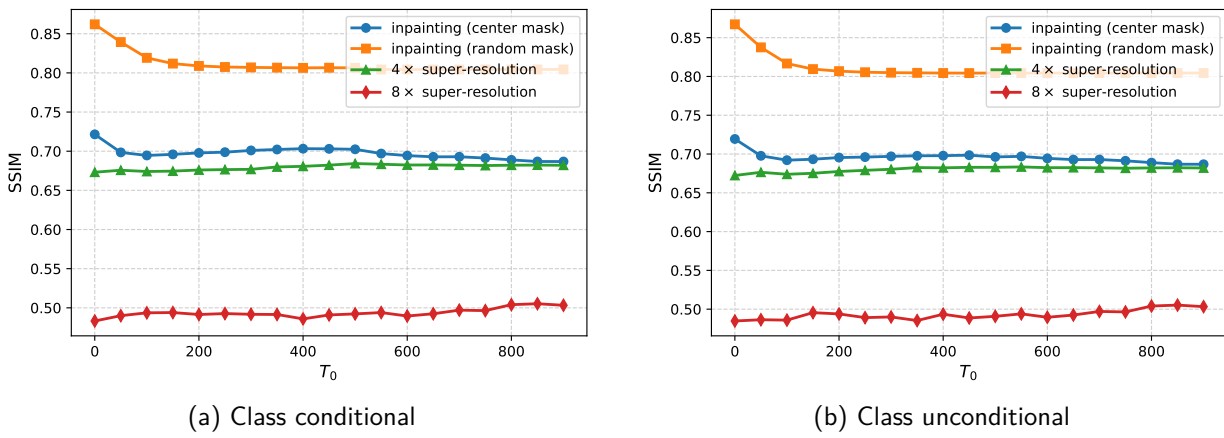

(a) Class conditional  (b) Class unconditional

Figure 7: Average SSIM scores across various inverse problems as a function of the guidance threshold $T_0$, which controls when each component of the piece-wise guidance term is active. $\Pi$GDM corresponds to $T_0 = 0$.

The simulations can be broadly categorized into two settings: class-conditional and class-unconditional. In the class-conditional case, it is assumed that the class label of the measurement image is known. A class-

---

[1]https://github.com/openai/guided-diffusion/blob/main/README.md

conditional diffusion model is employed, where the label is provided to the denoising process and incorporated during inference. In contrast, the class-unconditional case assumes that the class label is unknown at inference time; thus, the denoising process must operate solely based on the measurement without access to any class-specific information.

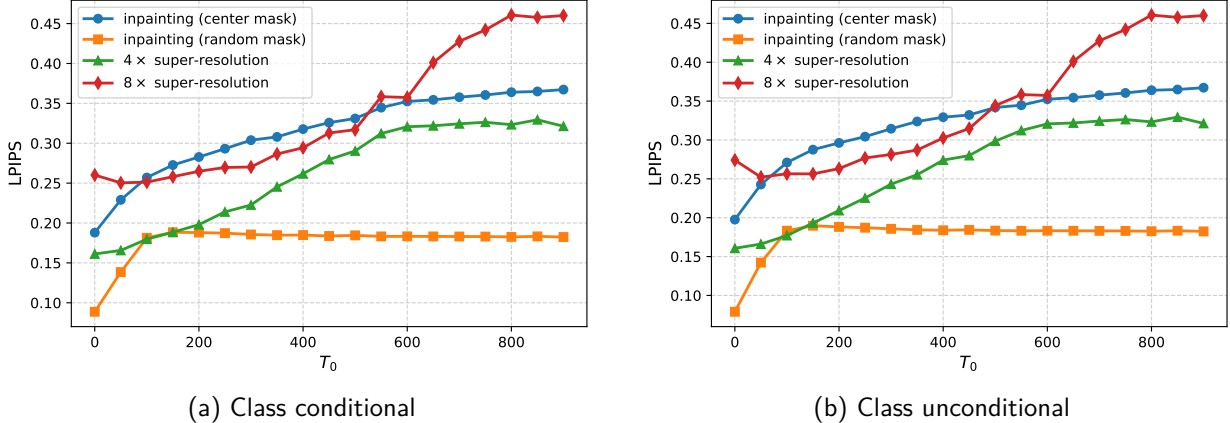

(a) Class conditional           (b) Class unconditional

Figure 8: Average LPIPS scores across various inverse problems as a function of the guidance threshold $T_0$, which controls when each component of the piece-wise guidance term is active. $\Pi$GDM corresponds to $T_0 = 0$.

For the inpainting task, two masking schemes are considered to assess robustness under structured and unstructured missing data. The first is a *central block mask*, where a contiguous $128 \times 128$ square region at the image center is missing. The second is a *random mask*, in which 30% of the pixels are randomly removed throughout the image. In the super-resolution setting, the images are downsampled using average pooling with scaling factors 4 and 8, aiming to reconstruct the original high resolution images from their low resolution counterparts.

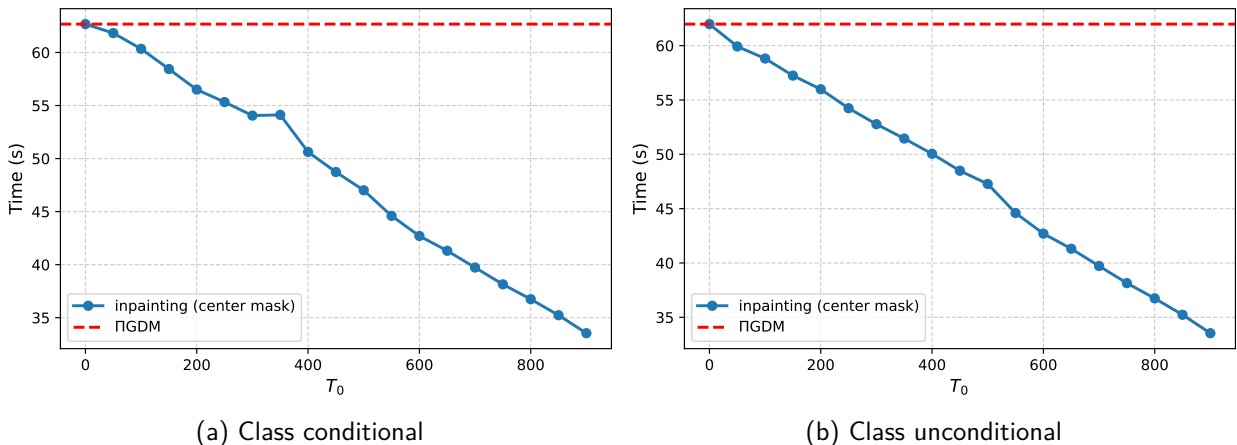

(a) Class conditional           (b) Class unconditional

Figure 9: Average inference time per image as a function of the guidance threshold $T_0$ for the inpainting task with a center mask. $T_0$ controls when each component of the piece-wise guidance term is active. $\Pi$GDM corresponds to $T_0 = 0$.

We evaluate performance using both pixel-level and perceptual metrics. Specifically, we report PSNR, SSIM, and learned perceptual image patch similarity (LPIPS). PSNR is a standard measure of reconstruction fidelity based on the pixel-wise error. SSIM, measures perceptual similarity by considering aspects such as luminance and contrast to follow how humans perceive an image, with values closer to one indicating

Table 2: Quantitative comparison across four inverse problems. Higher is better for PSNR/SSIM; lower is better for LPIPS and inference time. $T_0$ in our method is 200

| Inverse problem | PSNR ↑ | | SSIM ↑ | | LPIPS ↓ | | Time/img (s) ↓ | |
|---|---|---|---|---|---|---|---|---|
| | ΠGDM | Ours | ΠGDM | Ours | ΠGDM | Ours | ΠGDM | Ours |
| Inpainting (center mask) | 18.31 | 18.45 | 0.69 | 0.67 | 0.23 | 0.31 | 58.0 | 53.0 |
| Inpainting (random mask) | 27.73 | 26.57 | 0.84 | 0.78 | 0.10 | 0.19 | 58.2 | 53.1 |
| Super-resolution ×4 | 23.09 | 23.07 | 0.64 | 0.64 | 0.20 | 0.24 | 65.9 | 57.7 |
| Super-resolution ×8 | 20.09 | 20.00 | 0.46 | 0.47 | 0.30 | 0.30 | 58.8 | 53.9 |

better structural fidelity. LPIPS is also a perceptual metric and it leverages deep neural network features to evaluate perceptual similarity, where lower scores imply higher perceptual quality.

Figure 1 illustrates the behavior of the coefficient $\frac{1-\bar{\alpha}_t}{\bar{\alpha}_t}$ across diffusion time steps. As established in Theorems 1 and 2, the rapid decay of this coefficient at lower diffusion time-steps leads to a reduction in the KL divergence between the true problem-specific score function and its approximation. This behavior explains why the approximation is particularly accurate at lower diffusion time-steps and motivates the piecewise treatment adopted in the proposed method.

Figures 2–5 present restored images across different values of $T_0$, illustrating the visual quality achieved by the proposed method. Compared to ΠGDM, the results show that high-quality reconstructions are maintained even as $T_0$ increases, which enables faster restoration by reducing the number of computationally expensive steps.

Figures 2 and 4 correspond to class-conditional models, while Figures 3 and 5 show class-unconditional results. As expected, class conditioning improves visual quality due to the added label information, but the proposed method performs robustly in both settings. These findings highlight the efficiency and adaptability of the proposed method across a range of different inverse problems and diffusion model configuration.

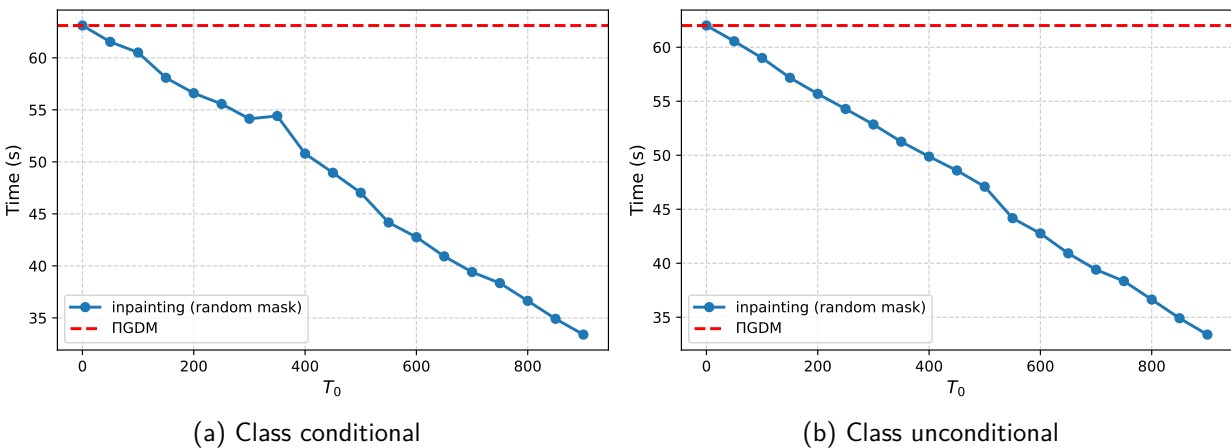

(a) Class conditional
(b) Class unconditional

Figure 10: Average inference time per image as a function of the guidance threshold $T_0$ for the inpainting task with a random mask. $T_0$ controls when each component of the piece-wise guidance term is active. ΠGDM corresponds to $T_0 = 0$.

Figure 6 presents the PSNR values across different inverse problems for varying values of $T_0$. The reported results correspond to the average PSNR computed over all images in the dataset, with the baseline ΠGDM represented by the data points at $T_0 = 0$. The figure indicates that increasing $T_0$ does not lead to significant variations in PSNR across the considered inverse problems, suggesting that the reduced validity of the piecewise approximation at higher diffusion time steps has a negligible impact on reconstruction fidelity as

measured by PSNR. A comparison between Fig. 6a and Fig. 6b further shows that class-conditional diffusion models achieve slightly higher PSNR values than their class-unconditional counterparts.

Figure 7 presents the SSIM metric for the same set of inverse problems evaluated at varying values of $T_0$. Higher SSIM values indicate better perceptual quality. Similar to the PSNR trends, an increase in $T_0$ does not lead to significant variations in SSIM across the considered inverse problems, showing that computational savings can be realized without significantly compromising the SSIM metric. Among the evaluated tasks, $8\times$ super-resolution exhibits the weakest performance, reflecting its increased difficulty due to the higher number of missing pixels. A comparison between Figures 7a and 7b again confirms that class-conditional diffusion models slightly outperform class-unconditional ones in terms of perceptual similarity.

Figure 8 evaluates the performance of the proposed approach in terms of the LPIPS metric for restoration results obtained using both class-conditional and class-unconditional diffusion models across various inverse problems and values of $T_0$. Unlike pixel-based metrics, LPIPS compares feature representations extracted by a neural network. In this simulation, AlexNet is used to assess perceptual similarity. Lower LPIPS values indicate higher perceptual quality. Unlike PSNR and SSIM, increasing $T_0$ leads to a noticeable increase in LPIPS, reflecting a degradation in image quality due to the diminished validity of the guidance term at higher time steps. However, the loss in performance remains minimal when $T_0$ is increased to 200, enabling substantial computational savings corresponding to 200 steps of faster score estimation. Among the tasks, $8\times$ super-resolution and inpainting with center mask exhibits the

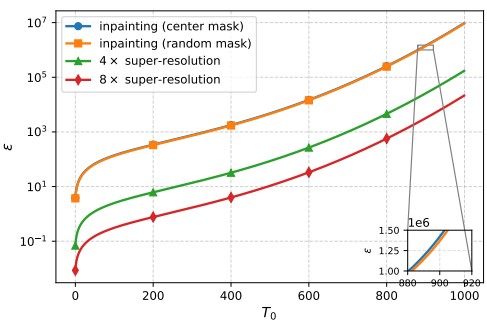

Figure 11: The score function approximation error curves in each inverse problem as a function of $T_0$, as characterized by Theorem 3.

greatest performance decline, whereas inpainting with a random mask shows the least. Similar to the cases of PSNR and SSIM, the class-conditional diffusion model slightly outperforms its class-unconditional counterpart according to the LPIPS metric.

Table 2 presents a quantitative comparison between ΠGDM and the proposed method across four inverse problems, evaluated on a 1000-image ImageNet subset. The proposed approach consistently reduces inference time per image by approximately 5–8 seconds across all tasks, while maintaining comparable reconstruction quality. These results demonstrate the effectiveness of the proposed method with $T_0 = 200$.

Figure 11 illustrates the score function estimation error $\epsilon$ for each inverse problem as a function of $T_0$, as characterized by Theorem 3. As $T_0$ increases, the estimation error of the score function increases accordingly. The figure also shows that different inverse problems exhibit distinct error curves, reflecting the influence of their respective degradation matrices $\boldsymbol{C}$. Moreover, the acceptable level of estimation error varies across problems. For example, as indicated by the LPIPS results in Fig. 8a and Fig. 8b, inpainting with a random mask can operate with larger values of $T_0$ than $4\times$ super-resolution, despite exhibiting higher score estimation error. This observation highlights that, when using these curves to identify a suitable initial value for $T_0$, both the degradation matrix of the inverse problem and its acceptable error tolerance must be taken into account. Under these conditions, the error curves provide useful guidance for selecting an appropriate starting point in the optimization of $T_0$.

Figures 9–13 present the average inference time per image across various inverse problems and different values of $T_0$. Here, inference time refers to the duration required by the algorithm to restore a given image. In all these figures, the baseline inference time corresponds to the ΠGDM algorithm at $T_0 = 0$, represented by a dashed red line. From Figs. 9–13, we can observe that increasing the value of $T_0$ generally accelerates the algorithm. This is because a higher $T_0$ implies fewer diffusion steps requiring the derivative computation of the diffusion model, a computationally expensive operation involved in calculating the guidance term.

Figure 9 presents the inference time per image for the inpainting problem with a center mask, for both class-conditional and class-unconditional diffusion models. At $T_0 = 500$, where losses in PSNR and SSIM are negligible, the algorithm achieves speed improvements of 24% and 25% for the unconditional and conditional models, respectively. The difference in speedup arises because the class-conditional denoising model is more

complex, making backpropagation for guidance computation more demanding. Consequently, reducing the number of these backpropagations yields greater computational savings.

Figure 10 illustrates inference times for the inpainting problem with random mask under various $T_0$ values for both model types. Consistent with previous results, increasing $T_0$ accelerates the algorithm. Notably, at $T_0 = 500$, where metric losses remain negligible, inference time is again reduced by 24% and 25% for unconditional and conditional models, respectively.

Figure 12 demonstrates inference times for the 4×super-resolution problem. Similar trends are observed: higher $T_0$ values lead to faster inference. At $T_0 = 500$, where metric degradation is minimal, speed improvements over ΠGDM amount to 21% for the unconditional model and 23% for the conditional model.

Finally, Figure 13 presents inference times for the 8× super-resolution task. From these two figures, we again observe that increasing $T_0$ decreases inference time. At $T_0 = 500$, the method is faster than ΠGDM by 23% and 24% for the unconditional and conditional diffusion models, respectively.

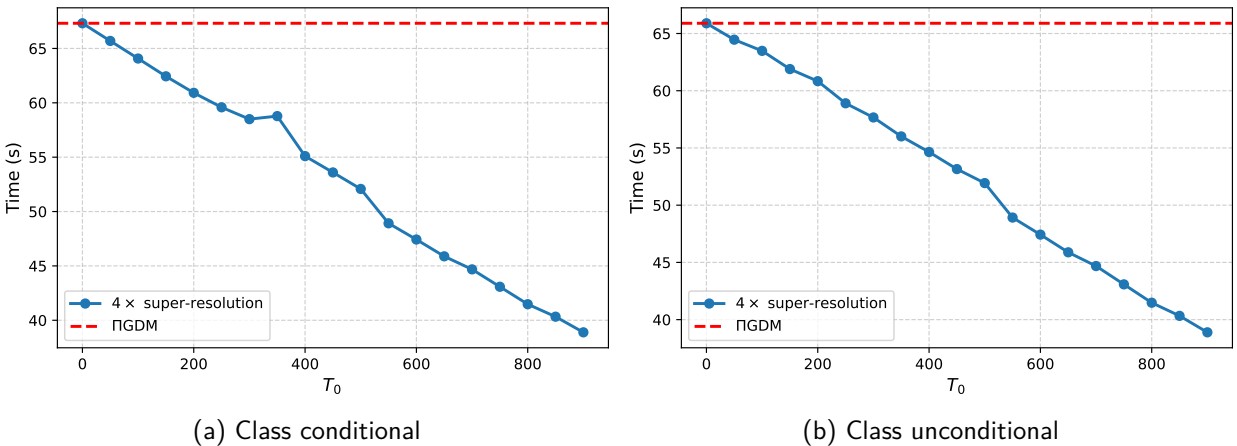

(a) Class conditional        (b) Class unconditional

Figure 12: Average inference time per image as a function of the guidance threshold $T_0$ for the 4× super-resolution task. $T_0$ controls when each component of the piece-wise guidance term is active. ΠGDM corresponds to $T_0 = 0$.

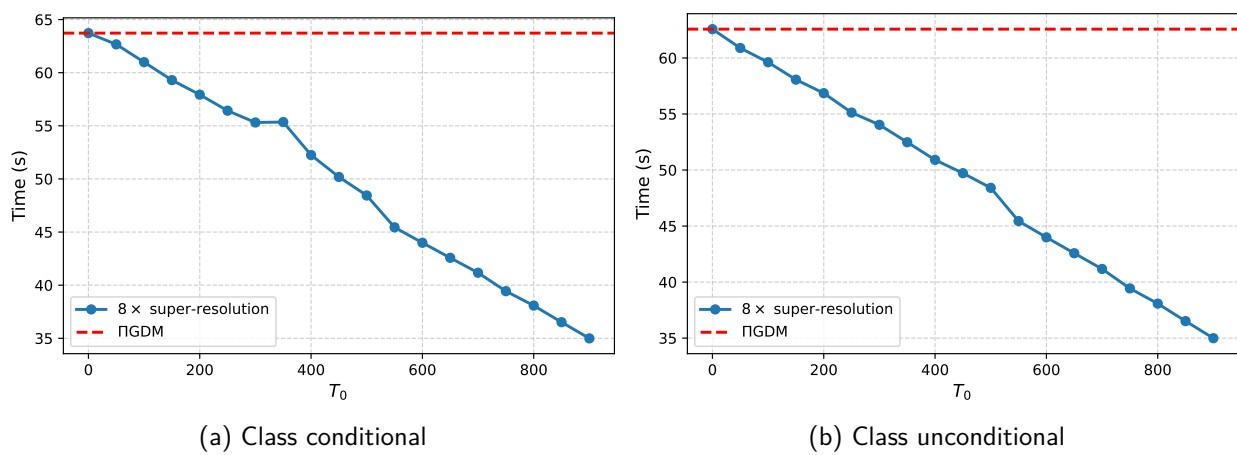

(a) Class conditional        (b) Class unconditional

Figure 13: Average inference time per image as a function of the guidance threshold $T_0$ for the 8× super-resolution task. $T_0$ controls when each component of the piece-wise guidance term is active. ΠGDM corresponds to $T_0 = 0$.

# 6 Conclusion

In this paper, a novel framework for solving inverse problems by posterior sampling using diffusion models has been developed. We have proposed a piecewise function approximation for the guidance term based on the different levels of diffusion noise determined by the time step. We have shown that in low time steps, the computation of the guidance term can be simplified to the gradient of a Gaussian function because the diffusion noise is insignificant in that regime. Our simulation results showed that an inference time reduction of at least 23% is achievable without loss of PSNR and SSIM in the restored images compared to the baseline ΠGDM.

### Acknowledgments

This research was supported by the U.S. National Science Foundation under Grant CNS-2210254.

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
