# OpenReview forum: "Diffusion Models for Solving Inverse Problems via Posterior Sampling with Piecewise Guidance"
_TMLR — Accepted by TMLR_

### Review · Reviewer_FxFE · 2025-12-23

**Summary Of Contributions:**

This paper presents a method for solving inverse problems using diffusion models, introducing a more efficient approximation for the problem-specific score, i.e., $\nabla x_t \log p(y \mid x_t)$, where $y$ is the measurement and $x_t$ is the (noisy) data at the $t$-th timestep of the forward diffusion process. This score is generally intractable since the distribution of $x_0 | x_t$ is intractable, where $x_0$ is the clean data. Observing that $x_t$ is obtained by adding Gaussian noise to $x_0$ with noise level increasing with $t$, the authors propose to treat $x_0$ as a deterministic function of $x_t$ at small timesteps $t < T_0$ (i.e., when the added noise is small), where $T_0$ is a tunable threshold. This leads to a simple approximation for $\nabla x_t \log p(y \mid x_t)$, which is more computationally efficient than conventional methods. For $t > T_0$, a conventional method [1] is used, making the overall approach a *piecewise* function of $t$. Theoretical analysis is conducted to characterize the approximaltion error and provide guidance on the choice of $T_0$. Experimental results demonstrate that the method accelerates inference while maintaining the reconstruction quality in terms of PSNR, SSIM, and LPIPS.

[1] Jiaming Song, Arash Vahdat, Morteza Mardani, and Jan Kautz. Pseudoinverse-Guided Diffusion Models for Inverse Problems. ICLR 2022.

**Audience:**

Yes

**Audience Explanation:**

The paper is relevant to people interested in improving the inference efficiency of diffusion models for inverse problems.

**Broader Impact Concerns:**

None.

**Claims And Evidence:**

No

**Claims Explanation:**

The proof of theoretical results seems incorrect.

**Requested Changes:**

My main question is regarding the proof. In a nutshell, the proof seems to treat two different quantities interchangeably. Specifically,

Theorem 1 analyzes:
$$
||\nabla_{x_t} \log p(Y = y \mid X_t = x_t, V_t = v_t) - \hat{F}(x_t, y)||_2^2,
$$

whereas Theorem 3 requires a bound on the following quantity:
$$
\mathbb{E}[ ||\nabla_{x_t} \log p(Y = y \mid X_t = x_t) - \hat{F}(x_t, y) ||_2^2 \mid V_t = v_t].
$$

Here, I use $\hat{F}(x_t, y)$ to denote the proposed approximation for $\nabla_{x_t} \log p(Y = y \mid X_t = x_t)$, and $V_t$ is the added noise at timestep $t$ as in the paper.

Notably, these two quantities are different due to the location of $V_t$ conditioning. However, (21) in the proof of Theorem 3 treats them interchangeably.


To elaborate, Theorem 3 uses the following equality to analyze the approximation error:
$$
\mathbb{E} [||\nabla_{x_t} \log p(Y = y \mid X_t = x_t) - \hat{F}(x_t, y)||^2 ] =
\mathbb{E}_{V_t} [ A ],
$$

where
$$
A := \mathbb{E} [||\nabla_{x_t} \log p(Y = y \mid X_t = x_t) - \hat{F}(x_t, y)||^2 \mid V_t ] .
$$

However, instead of analyzing $A$, Theorem 1 shows the following equality:
$$
B := || \nabla_{x_t} \log p(Y = y \mid X_t = x_t, V_t = v_t) - \hat{F}(x_t, y) ||_2 = \frac{\sqrt{1 - \bar{\alpha}_t}}{ \bar{\alpha}_t } ||C^T C v_t ||_2,
$$

using the property that $p(Y = y \mid X_t = x_t, V_t = v_t)$ is a Gaussian and admits a closed-form expression for $\nabla_{x_t} \log p$.

Theorem 3 then uses $B$ in place of $A$ to bound the expected error, which appears incorrect. Can the authors clarify this point?


**Other comments:**

- Although the comparison of inference wall time is informative, the results could be affected by CPU and GPU loads. Can the authors provide another metric that's more stable and precise, for example, the FLOPs required for computing the problem-specific score?

- "Figure 6... From this figure, we observe that, increasing T0 generally leads to a decline in pixel-level reconstruction quality, due to the reduced validity of the piecewise approximation at higher diffusion time steps."
I'm not sure if this claim holds, since PSNR increases or stays constant with T0 in most curves.

**Summary:**

Overall, the paper is easy to follow and the proposed method appears effective in improving inference efficiency. However, the key step in the proof of main theoretical claims is questionable and needs justification.

---

> ### Author Response · Authors · 2026-01-15
>
> Thank you for your feedback.
>
> Regarding the theoretical result, as the reviewer correctly notes, Theorem 1 analyzes
> $
> d = \big\lVert \nabla_{\boldsymbol{x}_t} \log p(Y = y \mid X_t = x_t, V_t = v_t) - \hat{F}(y, x_t) \big\rVert_2
> $
> and provides a closed-form expression for this quantity. We have followed the same notation as the reviewer in this expression.
> The quantity $d$ is a function of the random variable $V_t$. At each diffusion step, when $y$ and $x_t$ are given, $V_t$ is the only remaining source of randomness. As the reviewer mention, in Theorem 3, we are finding the expected value of the quantity $B$ in reviewers notation.
>
> The other quantity ($A$)  that reviewer mentions that must be used for taking the expected value, i.e., the expression:
>
> \begin{equation}
>     \mathbb{E}[\||\nabla_{\boldsymbol{x}_t}\log p(Y = y | X_t = x_t) - \hat{F}(y, x_t)\||_2 | V_t=v_t],
> \end{equation}
> is not a function of $v_t$ anymore and is constant with respect to that. To elaborate, $p(Y = y | X_t = x_t)$ is the result of following marginalisation i.e.
> \begin{equation}
>     p(Y = y | X_t = x_t) = \int p(Y = y | X_t = x_t, V_t = v_t)p(V_t = v_t| X_t = x_t) dv_t,
> \end{equation}
> and since based on the design of our method, $\hat{F}(y, x_t)$ is also not a function of $V_t$, then we have $\mathbb{E}[A] = A$. Although quantity $A$ can also be used as another metric, it is not what we want to bound. We want to calculate the error of the guidance term resulting from our approximation at each diffusion time step ($t$), which is a function of $V_t$ and bound the expected value of that.
>
> **Other comments**:
>
> * We were not able to report the achieved FLOPs for the operations due to hardware restrictions. While theoretical FLOPs can be estimated from hardware characteristics and runtime, this would not provide additional insight beyond the runtime results, so we report runtime only. We have nevertheless made our best effort to ensure that all experiments were conducted under the same conditions.
> * We extended the experiments by increasing $T_0$ up to 900 and updated the plots to report PSNR, SSIM, and LPIPS across all inverse problems. While the behavior of these metrics varies across tasks, the results consistently show that changes in PSNR and SSIM are negligible across different values of $T_0$, whereas LPIPS exhibits a clear degradation in perceptual quality as $T_0$ increases. We have modified the corresponding statements in the revised manuscript.

---

> > ### Comment · Reviewer_FxFE · 2026-01-21
> >
> > Dear authors,
> >
> > Thank you very much for your response!
> >
> > Could you please clarify why the quantity
> > \begin{equation} \mathbb{E}[||\nabla_{\boldsymbol{x}_t}\log p(Y = y | X_t = x_t) - \hat{F}(y, x_t)||_2 | V_t=v_t] \end{equation}
> > is not a function of $v_t$? This should be a function of $v_t$, since it is a conditional expectation conditioned on $V_t = v_t$.
> >
> > Furthermore, could you please clarify why the quantity is not relevant to the proof? In my understanding, the first equality in equation (21) uses this quantity, by using the following relation:
> > \begin{equation} \mathbb{E} [d^2] = \mathbb{E}_{v_t} [ \mathbb{E} [ ||\nabla_{\boldsymbol{x}_t}\log p(Y = y | X_t = x_t) - \hat{F}(y, x_t)||_2^2 | V_t=v_t ] ] . \end{equation}
> > If not, can the authors please justify the first equality in equation (21)?
> >
> > Additionally, could the authors clarify why the FLOPS could not be reported due to "hardware restrictions"?

---

> > > ### Author Response · Authors · 2026-01-22
> > >
> > > Thanks for your response.
> > >
> > > The reason why this quantity is not a function of $v_t$ is that both \begin{equation} \nabla_{\boldsymbol{x}_t}\log p(Y = y | X_t = x_t)
> > > \end{equation} and $\hat{F}(y, x_t)$ are not functions of $v_t$ therefore we have
> > >
> > > \begin{equation} \mathbb{E}[|\nabla_{\boldsymbol{x}_t}\log p(Y = y | X_t = x_t) - \hat{F}(y, x_t)|_2 | Vt=v_t] = , \end{equation}
> > >
> > > \begin{equation} |\nabla_{\boldsymbol{x}_t}\log p(Y = y | X_t = x_t) - \hat{F}(y, x_t)|_2, \end{equation}
> > >
> > > as $\hat{F}(y, x_t)$ does not depend on $v_t$ by its definition and $p(Y = y | X_t = x_t)$ is the result of the following marginalization which removes $v_t$ \begin{equation} p(Y = y | X_t = x_t) = \int p(Y = y | X_t = x_t, V_t = v_t)p(V_t = v_t| X_t = x_t) dv_t. \end{equation}
> > >
> > > In (21), we have the following \begin{equation} |\nabla_{\boldsymbol{x}_t}\log p(Y = y | X_t = x_t, V_t = v_t) - \hat{F}(y, x_t)|_2^2 = \frac{{1-\bar{\alpha}_t}}{{\bar{\alpha}_t}^2}|{\boldsymbol{C}^T\boldsymbol{C}\boldsymbol{v}_t}|_2^2. \end{equation} This is reflected in the theorem statement that we define $p_0(y|x_t)$ as the true conditional distribution, the one that $V_t$ is known for, i.e. $p(Y = y | X_t = x_t, V_t = v_t)$. Then we have calculated
> > >
> > > \begin{equation} \mathbb{E}[|\nabla_{\boldsymbol{x}_t}\log p(Y = y | X_t = x_t, V_t = v_t) - \hat{F}(y, x_t)|_2 ] \end{equation} With respect to $v_t$, to find the expected value of the error of our estimation.
> > >
> > > For the simulations in this manuscript, experiments were conducted on a shared external HPC cluster. Reporting achieved FLOPs would require access to low-level hardware performance counters, which are not available to users on this system due to administrative restrictions.

---

### Review · Reviewer_AECt · 2025-12-27

**Summary Of Contributions:**

The main contribution of the paper is a piecewise guidance scheme for solving inverse problems with pretrained diffusion priors. This guidance scheme operates in two phases when performing diffusion sampling from T to 0: in the first phase up till some timestep T_0, Pi-GDM updates are performed. After T_0 in the low noise regime, a new approximation is developed. Intuitively, this approximation uses the forward model p(y | x_0) as a direct proxy for p(y | x_t) since x_t = x_0 + noise is roughly x_0 when noise is small. Then, a theoretical analysis is performed that shows for a fixed time t, the KL divergence between the true conditional distribution p(y | x_t) at any fixed time t and proposed approximation can be bounded. With this bound, an exact setting of T_0 can be chosen to provide a certain approximation error epsilon. On an ImageNet subset of 50 images, the proposed method reduces inference time by about 25% while maintaining roughly similar accuracy to a baseline method, Pi-GDM.

Strengths:
1.	A simple piecewise guidance mechanism is proposed based on a nice observation of how diffusion iterates behave at low noise.

2.	A theoretical analysis provides some insights into the approximation error at varying time T.

3.	The method reduces inference time by about 25% which can be significant in practical applications.

Weaknesses (elaborated on in upcoming sections):
1.	The theoretical results do not have a clear tie to how the method is used in practice.

2.	Certain claims are not properly supported by the experiments and discussion of each experiment and its significance/relevance to the paper is missing.

**Audience:**

Yes

**Audience Explanation:**

The idea to use a very simple approximation at low noise levels is interesting and a practically good way to significantly reduce inference time. I believe some individuals in audience may find it interesting to tradeoff quality for inference time in this way.

**Claims And Evidence:**

No

**Claims Explanation:**

One claim is that the proposed method adapts across different diffusion model configurations, but the evidence towards that claim is only for class-unconditional vs conditional experiments. These experiments do not seem to add much value since introducing class conditioning changes very little in the architecture. Further, in figures like Figure 10, figure 6, figure 7, there is little to no difference between unconditional and conditional models so the added complexity in figures seems unnecessary. Is there a reason I am missing as to why these figures warrant both unconditional and conditional diffusion model results? A more convincing experiment towards the claim of adapting across different configurations would be varying the architecture (UNet vs transformer for example). Alternatively, this claim is not essential to the contributions of the paper.

If I understand Figure 6 correctly, it seems across almost all tasks besides inpainting (random mask), PSNR increases as T_0 increases which seems contradictory to the claim in the text that increasing T_0 leads to a decrease in pixel-level reconstruction quality. If this is with 1000 steps of diffusion sampling, then it would be helpful to see the limit as T_0 0 -> 1000 to see the hypothesized decline in reconstruction quality. Does it sharply drop off because the current plots show a very gradual change in perceptual quality (and that too, an increase?). Figure 8 seems to show the correct trend as LPIPS is increasing as T_0 increases, but Figure 6 seem incorrect. Similarly, the trend with SSIM is confusing in Figure 7 as for certain tasks like 4x SR, SSIM increases as T_0 increases, contradicting the claims in the text.

Figures 9-12 seem redundant to me which detracts from the clarity of the paper. The difference in PGDM and the proposed method is that a jacobian vector product and a matrix inverse is skipped in the first T_0 steps in the proposed method. Pi-GDM already proposes a rather efficient way to compute the matrix inverse for different forward operators, so the main inference time reduction should come from eliminating the jacobian-vector product. As such, varying the forward operator should not affect the inference time difference between the proposed method and Pi-GDM otherwise. Indeed, the trends of T_0 vs time are basically identical between Figure 9, Figure 10, Figure 11, and Figure 12. If the claim is to highlight that inference time decrease as function of T, it seems to me any one of these figures would suffice to make the claim. Is there a reason why the extra 4 figures are needed? Further, increasing T_0 should strictly make inference time reduce. Figures 9-12 should likely be averaged over several runs in which case probably the line should look closer to linear and monotonically decreasing.

One of the claimed contributions is that the proposed approach explicitly accounts for measurement noise. It is not clear how the approach does this because then T_0 should be a function of the measurement noise in either Theorem 3 or in the experiments.  Further, I believe Pi-GDM, DPS or DDRM already explicitly account for measurement noise in their approximations so it is incorrect to claim novelty along this axis.

For the experiments to back up the claims in the paper, I believe experiments need to be run for more than 50 images from Imagenet. Imagenet has 1000 classes so it is not clear which classes were chosen for the experiments. While not all experiments need to be run on the whole dataset, for at least one choice of T_0, I believe a standard Imagenet-validation dataset of 1000 images should be tested to back up the claim that the method preserves accuracy relative to Pi-GDM.

**Requested Changes:**

Major concerns (critical to securing recommendation for acceptance):

- In theorem 3, authors should discuss how the bound behaves as a function of delta. What does delta correspond to intuitively?  How does the term ||C^T C ||_F provide the “characteristics of the problem” – it is unclear what that means.
- the theoretical results should be tied to the practical method. More specifically, can T_0 be chosen using the setting from Theorem 3 – delta should be computable given C, so is T_0 not explicit for any given C and choice of epsilon?
- How is r_t chosen for timesteps after T_0?

Minor concerns (simply strengthen the work):

- Above equation 5: Unfortuantely -> unfortunately. Interactable -> intractable.
- Below equation 6: interactable -> intractable.
- Above theorem 3: conditinal -> conditional
- Figures are spread out across the paper. Figure 2 is on page 7 but not referenced till page 15.
- Overloaded notation: epsilon is denoiser error in theorem 2 but guidance error in theorem 3.

---

> ### Author Response · Authors · 2026-01-19
>
> Thank you for your feedback.
>
> The claim regarding adaptability across different diffusion models is grounded in the theoretical formulation of the proposed method. Specifically, the proposed score computation does not depend on the internal architecture of the diffusion model. For diffusion time steps lower than $T_0$, the guidance term is constructed in a manner that does not rely on the diffusion noise or on the specific parameterization of the score network. Instead, it is computed directly from the observed measurement $y$ and the noisy latent variable $x_t$. As a result, the method is inherently agnostic to the underlying diffusion model architecture (UNet, transformer, …) provided that the model supplies samples of $x_t$. Although conditional and unconditional models share similar architectures, they induce different noise estimations through different training objectives, making this comparison a reasonable sanity check for configuration-level robustness.
>
> Following the reviewer’s suggestion, we extended the experiments by increasing $T_0$ up to 900 diffusion steps and updated the plots to report PSNR, SSIM, and LPIPS across all inverse problems. While the behavior of these metrics varies across tasks, the results consistently show that changes in PSNR and SSIM are negligible across different values of $T_0$, whereas LPIPS exhibits a clear degradation in perceptual quality as $T_0$ increases. We have modified the corresponding statements in the manuscript.
>
> The reviewer raises a valid point regarding the similarity of the inference-time trends across Figs.9–12. The primary purpose of these figures is not to demonstrate qualitatively different trends, but rather to illustrate the consistency of the inference-time reduction across different inverse problems and across both class-conditional and class-unconditional diffusion models. While the proposed method reduces inference time primarily by eliminating the Jacobian–vector product beyond $T_0$, the absolute inference time and the magnitude of the speedup depend on both the inverse problem and the complexity of the underlying diffusion model. In particular, as the diffusion model becomes more computationally demanding, such as in the class-conditional setting, the cost of computing the score function increases, and the benefits of skipping this computation become more pronounced. The separate figures, therefore, highlight how the runtime gains of the proposed method scale with model complexity and problem setting, even though the qualitative dependence on $T_0$ remains similar.
>
> We have refined the wording related to measurement noise to better reflect our original intent and to avoid any potential ambiguity. The revised text now clarifies that the proposed method incorporates measurement noise within its formulation, without implying that this aspect is unique to our approach.
>
> In response to the reviewer’s concern, we have expanded the experimental evaluation to include a larger dataset. Specifically, we conducted an additional experiment using 1000 images from ImageNet for a representative value of $T_0$, and report the corresponding quantitative results in a newly added table in the revised manuscript. The result shows the reduction in runtime with negligible loss in PSNR, SSIM and LPIPS. Furthermore, we extended the range of evaluated $T_0$ values in the existing experiments. While the original submission considered values of $T_0$ in the range [0,500], the revised version includes additional evaluations up to $T_0 = 900$, providing a more comprehensive analysis of the trade-off between computational efficiency and reconstruction quality.
>
> **Requested changes**
> * We added further discussion after Theorem 3 to explain the behavior of the bound as a function of δ, provide intuition for δ as a tolerated guidance error, and clarify how $\||C^TC\||_F$ reflects problem-specific characteristics.
> * We strengthened the connection between Theorem 3 and the practical method by adding a figure and discussion showing how the bound can be used to guide the selection of $T_0$ given $C$  and a desired error tolerance.
> * $r_t$ is only used at the beginning of the backward path when the ΠGDM approach is used for calculation of the score, and $r_t$ is chosen as their own recommendation. We have made this clear in the revised version.
>
> **Minor concerns**
> * We have corrected the spelling issues. Regarding the ϵ notation, we retained the existing convention: boldface $\boldsymbol{\epsilon}$ is used to denote diffusion noise, following standard practice in the diffusion literature, while the regular symbol ϵ is used in the theoretical analysis to represent a scalar quantity.

---

### Review · Reviewer_7Qjy · 2026-01-08

**Summary Of Contributions:**

This paper proposes a piecewise guidance scheme for solving inverse problems using diffusion models. The key idea is to use different approximations for the guidance term depending on the diffusion timestep: a computationally simpler approximation at low timesteps (where diffusion noise is minimal) and the ΠGDM approach at higher timesteps. The authors demonstrate large inference time reduction compared to ΠGDM on image inpainting and super-resolution tasks, with minimal degradation in PSNR and SSIM metrics.

Strengths:
- Well-motivated approach that leverages the varying noise characteristics across diffusion timesteps
- Clear mathematical formulation with theoretical analysis
- Achieves meaningful computational savings with minimal quality loss
- Comprehensive experiments across multiple inverse problem settings

Weaknesses:
- Limited novelty, which essentially combines two existing methods in a straightforward piecewise manner
- Comparison only against one baseline (ΠGDM); missing comparisons with  and recent methods like DDRM, DDNM, DPS
- Relatively small-scale experiments
- Theoretical contributions are incremental applications of standard results

**Additional Comments:**

Minor issues:
- Abstract: "compared to the ΠGDM baseline" - define ΠGDM in the abstract or use full name
- P2: The transition from general inverse problems to diffusion models could be smoother
- Sec. 4: The theorem statements could be more formally presented with explicit assumption lists

**Audience:**

Yes

**Audience Explanation:**

The paper addresses a relevant problem - accelerating diffusion-based inverse problem solvers - which is of practical interest to the community. The large speedup with minimal quality loss represents a useful engineering contribution that practitioners working with diffusion models for inverse problems would find valuable.

However, the limited novelty and incremental nature of the contribution may reduce the broader interest. The work is essentially an optimization of an existing method (ΠGDM) rather than introducing new insights. The theoretical analysis, while correct, provides limited new insights beyond confirming the intuitive expectation that the approximation works better at low noise levels.

**Claims And Evidence:**

Yes

**Claims Explanation:**

The paper's main claims regarding computational speedup and maintained image quality are supported by experimental evidence. However, there are several concerns:

Supported claims:

- The large speedup over ΠGDM is clearly demonstrated across multiple tasks
- Minimal PSNR/SSIM degradation at T0=500 is shown empirically
- The theoretical analysis correctly derives KL divergence bounds

Insufficiently supported claims:

1. "Problem-agnostic framework": While tested on multiple tasks, the evaluation is limited to image restoration. Claims of generality would be stronger with diverse problem types (e.g., compressed sensing, medical imaging beyond the types tested).

2. "Explicitly accounts for measurement noise": This is presented as a novel contribution, but ΠGDM also incorporates measurement noise. The paper doesn't clearly differentiate its treatment from existing work.

3. Optimal T0 selection: Theorem 3 provides a criterion, but the paper doesn't validate whether this theoretical guidance aligns with empirically optimal T0 values. The experiments show results across different T0 but don't connect back to the theoretical predictions.


Experimental concerns:
- Only 50 images is a small sample size for drawing strong conclusions
- Visual quality differences in Figures 2-5 are subtle and subjective
- No worst-case analysis or failure case discussion

**Requested Changes:**

Critical changes:
1. Experimental comparisons:
    - Include recent diffusion-based inverse problem solvers as baselines (DDRM is discussed extensively but never compared)
    - Expand dataset to more images for statistical robustness
    - Add error bars and statistical significance tests to all quantitative plots
2. Fix technical errors:
    - Page 4: "interactable" → "intractable" (appears multiple times)
    - Equation (6): Clarify the conditioning - the current notation is confusing
3. T0 selection analysis:
    - Empirically validate Theorem 3's predictions: plot the theoretical bound vs. actual performance across T0 values
    - Provide clear guidelines for practitioners on how to select T0 for new problems
4. Novelty:
    - Tone down claims about "explicitly accounting for measurement noise" since ΠGDM also does this
    - Better differentiate from prior work in the introduction
    - Add a table clearly comparing the computational complexity of each method

Changes that would strengthen the work:

5. Theoretical analysis:
    - Provide convergence rate analysis
    - Discuss the fixed point of the piecewise method vs. ΠGDM
6. Experimental evaluation:
    - Test on larger images
    - Add memory consumption comparisons
    - Show wall-clock time vs. quality Pareto curves for different T0 values
    - Include ablation studies on the impact of different components
7. Presentation:
    - Figures 2-5: Add zoomed-in regions to highlight differences
    - Figure 1: Connect more explicitly to Theorems 1-2
    - Add a figure showing the guidance term magnitude across timesteps for both methods
    - Include visualizations of failure cases or challenging scenarios
8. Hyperparameter burden:
    - Compare the effort of tuning T0 vs. tuning other hyperparameters in baseline methods
    - Propose an adaptive scheme that automatically adjusts T0 during inference

---

> ### Author Response · Authors · 2026-01-19
>
> Thank you for your feedback.
>
> Response to reviewer 7Qjy (part 1/2)
>
> **Critical changes**
> 1. Experimental comparisons
>     * We acknowledge that other diffusion-based inverse problem solvers such as DDRM and DPS constitute important baselines. That said, our choice of ΠGDM as the primary baseline and only comparing to that is motivated by the following consideration. As reported in [1], ΠGDM consistently achieves improved reconstruction quality compared to DDRM and DPS across multiple inverse problems, albeit at the cost of increased computational complexity in comparison to DDRM. In our work, we demonstrate that our proposed method achieves performance comparable to ΠGDM while substantially reducing the computational cost through the proposed piecewise guidance strategy. As such, our results indirectly indicate that the proposed method would remain competitive with DDRM and DPS, while offering improved efficiency relative to a stronger baseline.
>    * In response to the reviewer’s concern, we have expanded the experimental evaluation to include a larger dataset. Specifically, we conducted an additional experiment using 1000 images from ImageNet for a representative value of $T_0$, and report the corresponding quantitative results in a newly added table in the revised manuscript. The result shows the reduction in runtime with negligible loss in PSNR, SSIM and LPIPS. Furthermore, we extended the range of evaluated $T_0$ values in the existing experiments. While the original submission considered values of $T_0$ in the range [0,500], the revised version includes additional evaluations up to $T_0=900$, providing a more comprehensive analysis of the trade-off between computational efficiency and reconstruction quality.
>    * In practice, due to the substantial computational cost, diffusion model evaluations are reported without error bars or statistical significance analysis, and it is common in the literature. This reporting convention is followed by many recent and widely cited works in diffusion-based inverse problems and image restoration [1,2,3].
> * Fix technical errors:
>   * The typo has been corrected throughout the manuscript. In addition, Equation (6) has been revised by rearranging the terms to improve readability and remove ambiguity.
> * $T_0$ selection analysis:
>   * Regarding empirical validation of the theoretical bound against the true score error, we note that this is inherently challenging, as the true score function is unknown in practice. Since the central objective of the method is precisely to obtain an accurate approximation of the score, direct evaluation of the theoretical bound against the true score is not feasible. For this reason, we rely on performance metrics and the derived theoretical curves to assess the practical implications of Theorem 3.
>   * In the revised manuscript, we have expanded the discussion following Theorem 3 to provide additional intuition regarding its implications. In addition, we have added theoretical error-level plots for different inverse problems in the simulation section, which illustrate how the bound in Theorem 3 varies with $T_0$ and depends on the degradation operator.
> We also include a clear discussion on how these theoretical curves can be used in practice to identify a suitable starting point for optimizing $T_0$, emphasizing that both the degradation matrix and the acceptable guidance error level must be taken into account when applying the theory to new problems.
> * Novelty:
>   * We have refined the wording related to measurement noise to better reflect our original intent and to avoid any potential ambiguity. The revised text now clarifies that the proposed method incorporates measurement noise within its formulation, without implying that this aspect is unique to our approach.
>   * To better differentiate our approach from existing methods, we have revised the introduction to place greater emphasis on the proposed piecewise guidance formulation.
>   * We added a table in section 3 that clearly presents a high level comparison of the computations required for calculation of the guidance term for each method.

---

> > ### Author Response · Authors · 2026-01-19
> >
> > Response to reviewer 7Qjy (part 2/2)
> >
> > **Changes that would strengthen the work**
> >
> > **Theoretical analysis**
> >
> > We note that providing convergence rate and fixed point analysis and providing them in detail requires the knowledge of the denoising model and makes the analysis short in scope. This falls out of the of the current work goal as we are providing a framework that works on as long as the denoising model is trained on the distribution of data $(x)$.
> >
> > **Experimental evaluation**
> >
> > We follow the common practice in the diffusion-based inverse problems literature and conduct all experiments on 256×256 images, which is the most widely reported resolution in prior work that enables meaningful comparisons [1][2]. We believe the chosen resolution is sufficient to demonstrate the effectiveness of the proposed approach.
> >
> > We already report PSNR, SSIM, and LPIPS across multiple inverse problems for a range of T_0values, together with the corresponding inference time per image. The information conveyed by additional time–quality curves would therefore be redundant with these results, while increasing the number of figures.
> >
> > Our method is governed by the parameter T_0, which determines the active regime of the piecewise formulation. When t>T_0, the method reduces to ΠGDM, whose parameters and ablation studies have already been thoroughly investigated in [1]. We believe that the current simulation result on the effect of T_0 and the ablation studies reported in [1] provides a comprehensive evaluation.
> >
> > **Presentation**
> >
> > We have added zoomed-in regions to Figs. 2–5 to better highlight qualitative differences between methods. In addition, we revised Fig.1 and its accompanying discussion to more explicitly connect the illustrated framework to Theorems 1 and 2, clarifying how the theoretical results motivate the proposed method. We have selected center-mask inpainting as a challenging representative case, since block removal is one of the most difficult image restoration settings. This task exhibits a wide range of reconstruction quality across methods and parameter choices, making it particularly informative about a challenging task.
> >
> > **Hyperparameter burden**
> >
> > We have added a discussion on how this parameter can be tuned, and how the provided theoretical insights can be used to guide its optimization more effectively across different inverse problems.
> >
> >
> > [1]: Song, Jiaming, et al. "Pseudoinverse-guided diffusion models for inverse problems." International Conference on Learning Representations. 2023.
> >
> > [2]: Kawar, Bahjat, et al. "Denoising diffusion restoration models." Advances in neural information processing systems 35 (2022): 23593-23606.
> >
> > [3]: Chung, Hyungjin, et al. "Improving diffusion models for inverse problems using manifold constraints." Advances in Neural Information Processing Systems 35 (2022): 25683-25696.